# Mitochondrial phenotypes in purified human immune cell subtypes and cell mixtures

**Shannon Rausser[1], Caroline Trumpff[1], Marlon A McGill[1], Alex Junker[1], Wei Wang[2], Siu-Hong Ho[2], Anika Mitchell[1], Kalpita R Karan[1], Catherine Monk[1,3,4], Suzanne C Segerstrom[5], Rebecca G Reed[6], Martin Picard[1,4,7]\***

[1]Department of Psychiatry, Division of Behavioral Medicine, Columbia University Irving Medical Center, New York, United States; [2]Columbia Center for Translational Immunology, Columbia University Irving Medical Center, New York, United States; [3]Department of Obstetrics and Gynecology, Columbia University Irving Medical Center, New York, United States; [4]New York State Psychiatric Institute, New York, United States; [5]Department of Psychology, University of Kentucky, Lexington, United States; [6]Department of Psychology, University of Pittsburgh, Pittsburgh, United States; [7]Department of Neurology, Merritt Center and Columbia Translational Neuroscience Initiative, Columbia University Irving Medical Center, New York, United States

**\*For correspondence:**
martin.picard@columbia.edu

**Competing interest:** The authors declare that no competing interests exist.

**Abstract** Using a high-throughput mitochondrial phenotyping platform to quantify multiple mitochondrial features among molecularly defined immune cell subtypes, we quantify the natural variation in mitochondrial DNA copy number (mtDNAcn), citrate synthase, and respiratory chain enzymatic activities in human neutrophils, monocytes, B cells, and naïve and memory T lymphocyte subtypes. In mixed peripheral blood mononuclear cells (PBMCs) from the same individuals, we show to what extent mitochondrial measures are confounded by both cell type distributions and contaminating platelets. Cell subtype-specific measures among women and men spanning four decades of life indicate potential age- and sex-related differences, including an age-related elevation in mtDNAcn, which are masked or blunted in mixed PBMCs. Finally, a proof-of-concept, repeated-measures study in a single individual validates cell type differences and also reveals week-to-week changes in mitochondrial activities. Larger studies are required to validate and mechanistically extend these findings. These mitochondrial phenotyping data build upon established immunometabolic differences among leukocyte subpopulations, and provide foundational quantitative knowledge to develop interpretable blood-based assays of mitochondrial health.

## Editor's evaluation

In this manuscript the authors have assessed age-, sex- and time-driven differences in mitochondrial phenotypes with human peripheral blood mononuclear cells and their subsets. They find that differences in metabolic profile can be driven by differences in leukocyte composition (driven by age, sex, other stimuli) and platelet contamination. The leads obtained from this study would be useful for further research.

## Introduction

Mitochondria are the most studied organelle across the biomedical sciences (*Picard et al., 2016*). The growing focus on mitochondria is motivated by evidence positioning mitochondrial (dys)function as a driver of disease risk and aging (*Jang et al., 2018*; *Picard et al., 2016*; *Wallace, 2015*), and as a

mediator of brain-body processes that shape health and disease trajectories across the lifespan (*Picard et al., 2019*). Addressing emerging biomedical research questions around the role of mitochondria on human health requires tractable quantitative biomarkers of mitochondrial content (the amount or mass of mitochondria per cell) and function (energy production capacity) that can be deployed in accessible human tissues, such as peripheral blood leukocytes. To develop such biomarkers, we need to establish standard effect sizes of mitochondrial variation. between immune cell subtypes and over time. and to quantitatively define potential technical confounds and covariates such as age, sex, and known biomarkers.

Although some cell-specific assays can interrogate immune cells' mitochondrial function with a reasonable degree of cell specificity (*Chacko et al., 2013*), more frequent approaches in the literature use peripheral blood mononuclear cells (PBMCs) (*Dixon et al., 2019*; *Ehinger et al., 2016*; *Karabatsiakis et al., 2014*; *Picard et al., 2018*; *Tyrrell et al., 2015*; *Weiss et al., 2015*). These approaches largely assume that the immunometabolic properties of different immune cells have a negligible influence on mitochondrial measurements. However, there are marked differences in the metabolic properties of different immune cell subtypes well known to immunologists. For example, lymphocytes and monocytes significantly differ in their respiratory properties and mitochondrial respiratory chain (RC) protein abundance (*Chacko et al., 2013*; *Kramer et al., 2014*; *Maianski et al., 2004*; *Pyle et al., 2010*). In various leukocyte subtypes, these divergent immunometabolic properties even contribute to determine the acquisition of specialized cellular characteristics (*Pearce et al., 2013*). The activation, proliferation, and differentiation of monocytes (*Nomura et al., 2016*) and T cells (*Michalek et al., 2011*) into specific effector cells require distinct metabolic profiles and cannot proceed without the proper metabolic states. Likewise, naïve and memory T lymphocytes differ in their reliance on mitochondrial oxidative phosphorylation (OxPhos) involving the RC enzymes (*Brand, 1985*; *Jones et al., 2019*; *Ron-Harel et al., 2019*), and harbor differences in protein composition and mitochondrial content within the cytoplasm (*Bektas et al., 2019*). Thus, the immune system offers a well-defined landscape of metabolic profiles which, if properly mapped, can potentially serve as biomarkers.

The composition of peripheral blood leukocytes in the human circulation is influenced by several factors. Immune cell subtypes are normally mobilized from lymphoid organs into circulation in a diurnal fashion and with acute stress (*Ackermann et al., 2012*; *Beis et al., 2018*; *Dhabhar et al., 2012*; *Dhabhar et al., 1994*). The abundance of circulating immune cell subtypes also vary extensively between individuals, partially attributable to both individual-level (e.g., sex and age) and environmental factors (*Patin et al., 2018*). As a result, sampling whole blood or mixed PBMCs from different individuals may reflect different cell populations that would therefore not be directly comparable.

Furthermore, frequently used Ficoll-isolated PBMCs are naturally contaminated with (sticky) platelets (*Butler et al., 2007*). Platelets contain mitochondria and mtDNA but no nuclear genome to use as reference for mtDNA copy number (mtDNAcn) measurements (*Hurtado-Roca et al., 2016*), representing a major source of bias to mitochondrial studies in PBMCs, buffy coat, or other undefined cell mixtures (*Banas et al., 2004*; *Shim et al., 2020*; *Urata et al., 2008*). But the extent to which platelet contamination influences specific mitochondrial content and RC capacity features in PBMCs has not been quantitatively defined.

Another significant gap in knowledge relates to the natural dynamic variation in mitochondrial content and function over time. Mitochondria dynamically recalibrate their form and functions in response to exercise (*Gan et al., 2018*) and mental stress (*Picard and McEwen, 2018*). Mitochondria also contain receptors that enable their functions to be tuned by humoral metabolic and endocrine inputs (*Bénard et al., 2012*; *Du et al., 2009*). Thus, cell-specific mitochondrial features could vary over time. To develop valid blood-based mitochondrial markers, we therefore need to determine whether leukocyte mitochondria are stable *trait*-like properties of each person, or *state*-like properties possibly varying in response to metabolic or endocrine factors.

To address these questions, we used a high-throughput mitochondrial phenotyping platform on immunologically defined immune cell subtypes, in parallel with PBMCs, to quantify cell subtype-specific mitochondrial phenotypes in a small, diverse cohort of healthy adults. First, we establish the extent to which cell type composition and platelet contamination influence PBMC-based mitochondrial measures. We then systematically map the mitochondrial properties of different immune cell subtypes and validate the existence of stable mitochondrial phenotypes in an intensive repeated

measures design within the same individual, which also begins to reveal a surprising degree of intra-individual variation over time. Collectively, these data confirm and quantify the biological limitations of PBMCs to profile human mitochondria, introduce the concept of multivariate mitotypes, and define unique cell-specific mitochondrial features and their effect sizes in circulating human leukocytes in relation to age, sex, and biomarkers that can guide future studies. Taken together, these data represent a resource to design cell-specific immune mitochondrial phenotyping strategies.

## Results

### Cell subtype distributions by age and sex

We performed mitochondrial profiling on molecularly defined subtypes of immune cell populations in parallel with PBMCs in 21 participants (11 women, 10 men) distributed across four decades of life (ages 20–59, 4–8 participants per decade across both sexes). From each participant, 100 ml of blood was collected; total leukocytes were then labeled with two cell surface marker cocktails, counted, and isolated by fluorescence-activated cell sorting (FACS; see Materials and methods for details), frozen, and subsequently processed as a single batch on our mitochondrial phenotyping platform adapted from *Picard et al., 2018* (*Figure 1a*). In parallel, a complete blood count (CBC) for the major leukocyte populations in whole blood, standard blood chemistry, and a metabolic and endocrine panel were assessed (*Figure 1b*).

We first quantified the abundance of specific cell subtypes based on cell surface marker combinations (*Figure 1c*, *Figure 1—figure supplements 1–2*, and *Supplementary file 1*). Men had 66% fewer CD4+ naïve T cells than women (p<0.05) and tended to have on average 35–44% more NK cells and monocytes (*Figure 1d*). These differences were characterized by moderate to large standardized effect sizes (Hedge's g=0.71–0.93), consistent with recent findings (*Márquez et al., 2020*). Between individuals of the same sex, the circulating proportions of various cell subtypes (e.g., B cells range: <0.01–15.3%, see *Supplementary file 2*) varied by up to an order of magnitude (i.e., 1tenfold) (*Figure 1e*).

In relation to age, as expected (*Patin et al., 2018*), CD8+ naïve T cell abundance was lower in older individuals (p<0.01). Compared to young adults in their 20s, middle-aged individuals in their 50s had on average ~63% fewer CD8+ naïve T cells (*Figure 1f*). In contrast, effector memory CD4+ (CD4+ EM) and central memory CD8+ (CD8+ CM) cell abundance tended to increase with age (positive correlation, r=0.31 for both), an overall picture consistent with immunological aging (*Márquez et al., 2020*; *Nikolich-Žugich, 2014*; *Patin et al., 2018*).

CBC-derived cell proportions also showed that men had on average 28% more monocytes than women (*Figure 1—figure supplement 3*), consistent with our FACS results. Conversely, women had on average 20% more platelets than men. Platelet abundance also tended to decrease with age, a point discussed later.

### Circulating cell composition influence PBMCs mitochondrial phenotypes

We next examined how much the abundance of various circulating immune cell subtypes correlated with individual mitochondrial metrics in PBMCs. Our analysis focused on two key aspects of mitochondrial biology: (i) *mitochondrial content*, indexed by citrate synthase (CS) activity, a Kreb's cycle enzyme used as a marker of mitochondrial volume density (*Larsen et al., 2012*), and mtDNAcn, reflecting the number of mtDNA copies per cell; and (ii) *RC function* measured by complex I (CI), complex II (CII), and complex IV (CIV) enzymatic activities, which reflect the capacity for electron transport and respiratory capacity and serve here as a proxy for maximal RC capacity. Furthermore, by adding the three mean-centered features of RC function together as a numerator (CI+CII+ CIV), and dividing this by the combination of content features (CS+mtDNAcn), we obtained an index reflecting *RC capacity on a per-mitochondrion basis*, known as the mitochondrial health index (MHI) adapted from previous work (*Picard et al., 2018*).

As expected, the abundance of multiple circulating cells was correlated with PBMCs mitochondrial features (*Figure 2*). Notably, the correlation between circulating B cell abundance and PBMCs CS activity was r=0.78 (p<0.0001), meaning that the proportion of shared variance ($r^2$) between both variables is 61% (i.e., B cell abundance explains 61% of the variance in PBMCs CS). Similarly, the correlations between B cell abundance and PBMCs mtDNAcn, CI, CII activities ranged from r=0.52

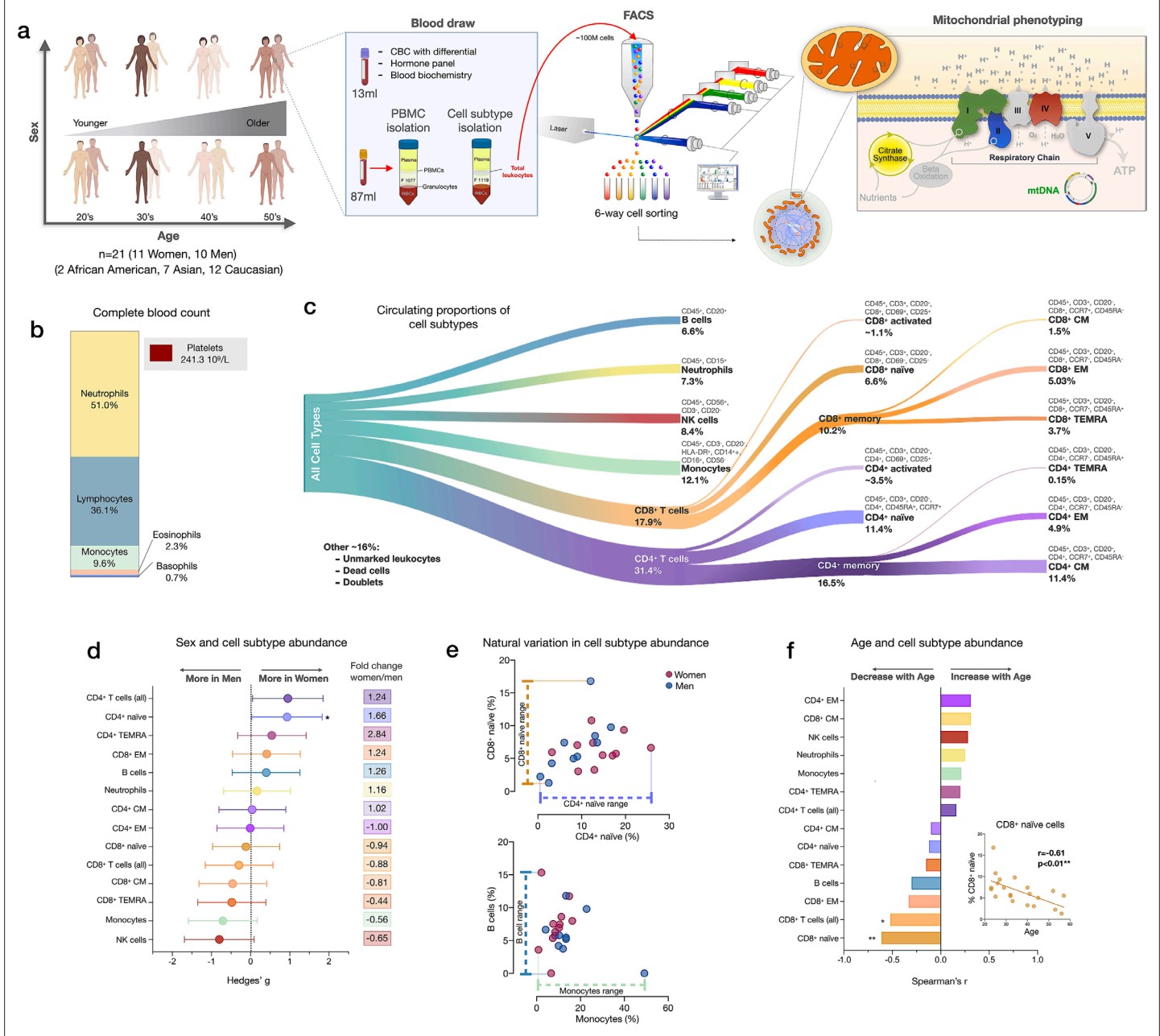

**Figure 1.** Immune cell subtype distribution in adult women and men. (**a**) Overview of participant demographics, blood collection, processing, and analysis pipeline. Total leukocytes were isolated using Ficoll 1119 and PBMCs were isolated on Ficoll 1077. (*right*) The five mitochondrial features analyzed on the mitochondrial phenotyping platform are colored. Five million cells per purified cell subtype were used for analyses. Mitochondrial phenotyping platform schematic adapted from **Figure 1a** in *Picard et al., 2018*. (**b**) Stacked histogram showing the leukocytes distribution derived from the complete blood count (CBC) of major cell types. (**c**) Diagram illustrating the proportion of circulating immune cell subtypes (% of all detected cells) quantified by flow cytometry from total peripheral blood leukocytes. Cell surface markers and subtype definitions are detailed in ***Supplementary file 1***. (**d**) Effect sizes for cell subtype distribution differences between women (n=11) and men (n = 10). p-values from nonparametric Mann-Whitney t-test. Error bars reflect the 95% confidence interval (CI) on the effect size, and the fold change comparing raw counts between women and men is shown on the right. (**e**) Example distributions of cell type proportions in women and men illustrating the range of CD4⁺ and CD8⁺ naïve cells, B cells, and monocytes, highlighting the natural variation among our cohort. Each data point reflects a different individual. (**f**) Spearman's r correlation between age and cell types proportion. n=21, p<0.05*, p<0.01**. PBMC, peripheral blood mononuclear cell.

The online version of this article includes the following figure supplement(s) for figure 1:

**Source data 1.** Immune cell subtype distribution in adult women and men.

**Figure supplement 1.** Diagram of leukocyte cell lineages.

*Figure 1 continued on next page*

*Figure 1 continued*

**Figure supplement 2.** Gating strategy to quantify all cell subtypes and sorting major cell subtypes for mitochondrial phenotyping.

**Figure supplement 3.** Sex differences and age correlations with leukocyte abundance measured by complete blood count (CBC).

**Figure supplement 3—source data 1.** Sex differences and age correlations with leukocyte abundance measured by complete blood count.

to 0.67, ps<0.05–0.01 (27–45% of shared variance). The circulating proportions of other cell types accounted for more modest portions ( $r^2$<14%) of the variance in PBMCs, although the higher abundance of memory cells tended to be negatively associated with PBMC RC enzymatic activities.

Based on CBC-derived cell proportions, the abundance of eosinophils and neutrophils was positively correlated with most PBMC mitochondrial content and activity features (*Figure 2—figure supplement 1*). Because PBMCs do not contain granulocytes, these correlations may reflect the independent effect of a humoral factor on cell mobilization and mitochondrial function. Taken together, these data confirmed that mitochondrial features assessed in PBMCs in part reflect the proportions of some but not all circulating cell subtypes, quantitatively documenting how cell type distribution may confound the measurements of mitochondrial function in PBMCs.

## Platelets influence PBMCs mitochondrial phenotypes

Given that platelets easily form platelet-leukocyte aggregates (*Butler et al., 2007*; *Figure 3a*), to partly resolve the origin of the discrepancies between isolated cell subtypes and PBMCs noted above, we directly quantified the contribution of platelets to total mitochondrial content and activity features in PBMCs. We note that the PBMCs in our experiments were carefully prepared with two passive platelet depletion steps (low-speed centrifugations, see Appendix 1 for details), which should have already produced 'clean' PBMCs.

We first asked if the abundance of platelets from the CBC data in the cohort varies by age. Consistent with two large epidemiological studies of >40,000 individuals (*Biino et al., 2013*; *yong, 2015*), we found that platelet count decreased by ~ 6% for each decade of life (*Figure 3b–c*). This reflects a decline of 24% between the ages of 20 and 60, although the effect sizes vary by cohort and our estimate is likely overestimated due to the small size of our cohort. As expected, total platelet count tended to be consistently *positively* correlated with mtDNAcn, CS, and RC activities in PBMCs (r=0.031–0.38) (*Figure 3d*). Therefore, the age-related loss of platelets (and of the mtDNA contained within them) could account for the previously reported age-related decline in mtDNAcn from studies using either whole blood (*Mengel-From et al., 2014*; *Verhoeven et al., 2018*) (which includes all platelets) or PBMCs (*Zhang et al., 2017*) (which include fewer contaminating platelets).

We directly tested this hypothesis by immunodepleting platelets from 'clean' PBMCs and comparing three resulting fractions: total PBMCs, actively platelet-depleted PBMCs, and platelet-enriched eluate. As expected, platelet depletion decreased mtDNAcn, CS, and RC activities, indicating that contaminating platelets exaggerated specific mitochondrial features by 9–22%, except for complex IV (*Figure 3f–g*). Moreover, the platelet-enriched eluate showed 23–100% higher mitochondrial activities relative to total PBMCs, providing direct evidence that the active platelet depletion method was effective and that platelets inflate estimates of mitochondrial abundance and RC activity in standard PBMCs prepared with two passive platelet-depletion steps. Interestingly, the composite MHI was minimally affected by the platelet depletion procedure, suggesting that this multivariate index of RC capacity on a per-mitochondrion basis may be more robust to platelet contamination than its individual features.

## Individual cell subtypes are biologically distinct from PBMCs contaminated with platelets

Mitochondrial phenotyping was performed in FACS-purified immune cells, in parallel with PBMCs. To obtain sufficient numbers of cells for mitochondrial phenotyping, we selected the six most abundant cell subtypes for each individual and isolated 5×10⁶ cells for each sample. Because memory subtypes were relatively rare, CM and EM subtypes were pooled for CD4⁺ and/or CD8⁺ (CM-EM). This generated a total of 340 biological samples, including 136 biological replicates, yielding 204 individual participant/cell subtype combinations used in our analyses.

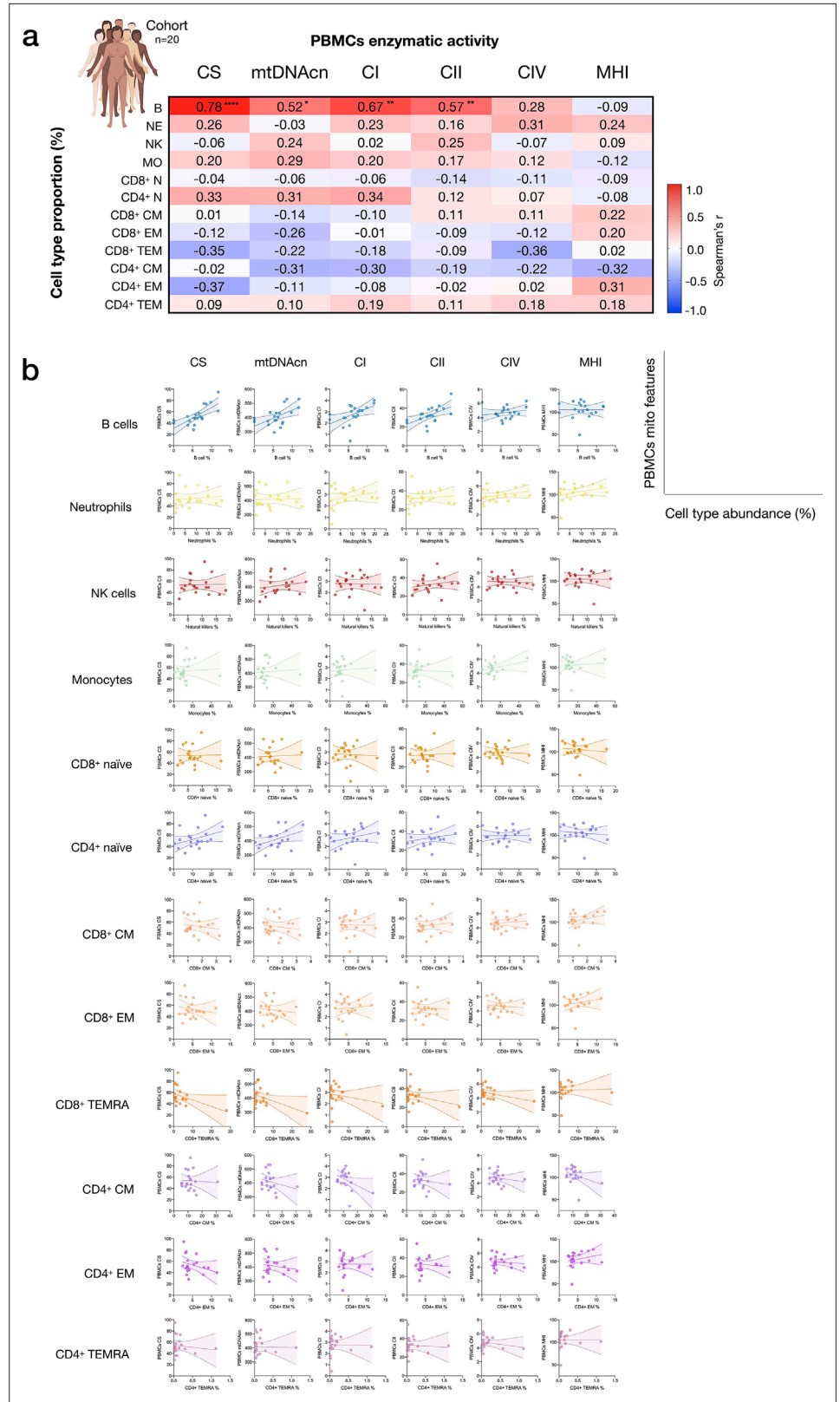

**Figure 2.** Influence of cell subtypes on mitochondrial features in total PBMCs. (**a**) Pairwise correlations (Spearman's r) between cell subtype proportions obtained from cell sorting with mitochondrial features measured in PBMCs for the cohort (n=20, one participant missing PBMC data). Aggregate correlations are shown as a heatmap (*top*) and (**b**) individual scatterplots (*bottom*). p<0.05*, p<0.01**, p<0.0001****. PBMC, peripheral blood mononuclear cell.

*Figure 2 continued on next page*

*Figure 2 continued*

The online version of this article includes the following figure supplement(s) for figure 2:

**Source data 1.** Influence of cell subtypes on mitochondrial features in total PBMCs.

**Figure supplement 1.** Associations between CBC cell proportions with mitochondrial features measured in PBMCs.

**Figure supplement 1—source data 1.** Associations between CBC cell proportions with mitochondrial features measured in PBMCs.

---

Among cell subtypes, CS activity was highest in monocytes and B cells, and lowest in CD4$^+$ naïve T cells, with other cell types exhibiting intermediate levels (*Figure 4a*). Regarding mitochondrial genome content, B cells had the highest mtDNAcn with an average of 451 copies per cell compared to neutrophils and NK cells, which contained only an average of 128 (g=5.94, p<0.0001) and 205 copies (g=3.84, p<0.0001) per cell, respectively (*Figure 4b*). Naïve and memory CD4$^+$ and CD8$^+$ T lymphocytes had intermediate mtDNAcn levels of ~300 copies per cell, except for CD8$^+$ naïve cells (average of 427 copies per cell). Between cell types, CS activity and mtDNAcn differed by up to 3.52-fold.

In relation to RC function, monocytes had the highest complex I, II, and IV activities. Consistent with their low mtDNAcn, neutrophils also had the lowest activities across complexes, whereas naïve and memory subtypes of T and B lymphocytes presented intermediate RC enzyme activities (*Figure 4c–e*). PBMCs had up to 2.9-fold higher levels of CS, CI, and CII activity per cell than any of the individual cell subtypes measured, again consistent with platelet contamination.

The correlations between different mitochondrial features indicate that CS and mtDNAcn were only weakly correlated with each other, and in some cases were negatively correlated (*Figure 4f*). This finding may be explained by the fact that although both CS and mtDNAcn are positively related to mitochondrial content, CS may be superior in some cases (*Larsen et al., 2012*), inadequate in specific tissues (*McLaughlin et al., 2020*), and that mtDNAcn can change independently of mitochondrial content and biogenesis (*Picard, 2021*). For RC complexes CI, CII, and CIV, which physically interact and whose function is synergistic within the inner mitochondrial membrane, correlations tended to be positive, as expected (*Figure 4f*). However, relatively weak and absent inter-correlations between mitochondrial features in some cell types reveal that each metric (i.e., content features and enzymatic activities) provides relatively independent information about the immune cell mitochondrial phenotype.

We extended the analyses of univariate metrics of mitochondrial content and function by exploring the multivariate MHI, which significantly differed between cell subtypes (p<0.0001) (*Appendix 2—figure 1*). These results are discussed in Appendix 2.

## Mitochondrial features exhibit differential co-regulation across immune cell subtypes

Next, we asked to what extent mitochondrial markers correlate across cell subtypes in the same person (co-regulation). For example, we determined whether having high mtDNAcn or low MHI could constitute coherent properties of an individual that are expressed ubiquitously across cell types (e.g., an individual with the highest mtDNAcn in B cells also having the highest mtDNAcn in all cell types), or if these properties are specific to each cell subtype.

CS activity and mtDNAcn were moderately co-regulated across cell subtypes (average correlation $r_{z'}$=0.63 and 0.53, respectively) (*Appendix 2—figure 1*). In comparison, RC enzymes showed markedly lower correlations between cell types and some cell types were not correlated with other cell types, revealing a substantially lower degree of co-regulation among RC components than in mitochondrial content features. MHI showed moderate and consistent positive co-regulation across cell types (average $r_{z'}$=0.37). Notably, PBMCs exhibited moderate to no correlation with other cell subtypes, further indicating their departure from purified subtypes (*Figure 4—figure supplement 1*). Taken together, these subtype-resolution results provide a strong rationale for performing cell type-specific studies when examining the influence of external exposures and person-level factors on immune cells' mitochondrial bioenergetics, including the influence of sex and age.

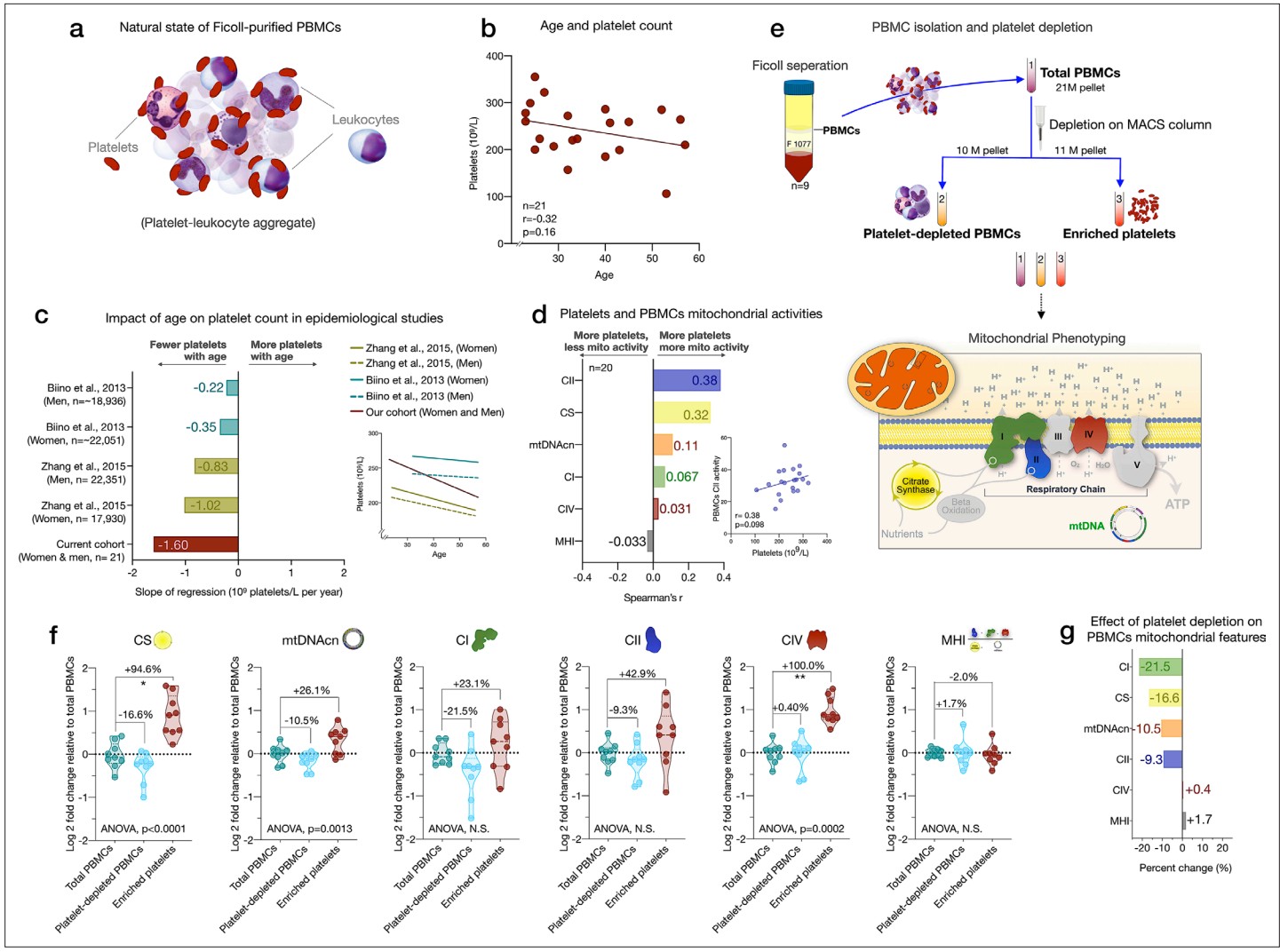

**Figure 3.** Influence of platelet contamination on mitochondrial features in total PBMCs. (**a**) Schematic of the natural state of Ficoll-isolated PBMCs associated with contaminating platelets. (**b**) Association of age and circulating platelet abundance (i.e., count) in our cohort (Spearman's r). (**c**) Change in platelet abundance as a function of age. The magnitude of the association (slope of the regression: 109 platelets/L per year) from two large epidemiological studies and our cohort. The inset shows the actual regressions (n=21–22,351). (**d**) Effect sizes of the association between platelet count and PBMC mitochondrial features in our cohort (n=20). (**e**) Overview of the experimental PBMC platelet depletion study, yielding three different samples subjected to mitochondrial phenotyping. Mitochondrial phenotyping platform schematic adapted from **Figure 1a**, *Picard et al., 2018*. (**f**) Fold change in mitochondrial parameters between (i) platelet-depleted PBMCs, and (ii) enriched platelets (with contaminating PBMCs), relative to (iii) total PBMCs. p-values from one-way nonparametric ANOVA Friedman test, post hoc Dunn's multiple comparisons relative to total PBMCs. (**g**) Percent change of platelet-depleted PBMCs mitochondrial features from total PBMCs. n=9, p<0.05*,p<0.01**, p<0.001***, p<0.0001****. PBMC, peripheral blood mononuclear cell.

The online version of this article includes the following figure supplement(s) for figure 3:

**Source data 1.** Influence of platelet contamination on mitochondrial features in total PBMCs.

## Mitochondrial content and RC function differ between women and men

To explore the added value of cell subtype specific studies when applied to real-world questions, we systematically compared CS activity, mtDNAcn, RC activity, and MHI between women and men (*Figure 5a–f*). Given the exploratory nature of these analyses with a small sample size, only two results reached statistical significance (analyses non-adjusted for multiple testing). Compared to men, women had 29% higher CS activity in CD8[+] CM-EM T cells (g=1.52, p<0.05, *Figure 5a*) and 26% higher CI activity in monocytes (g=1.35, p<0.05, *Figure 5c*). Interestingly, across all cell subtypes examined,

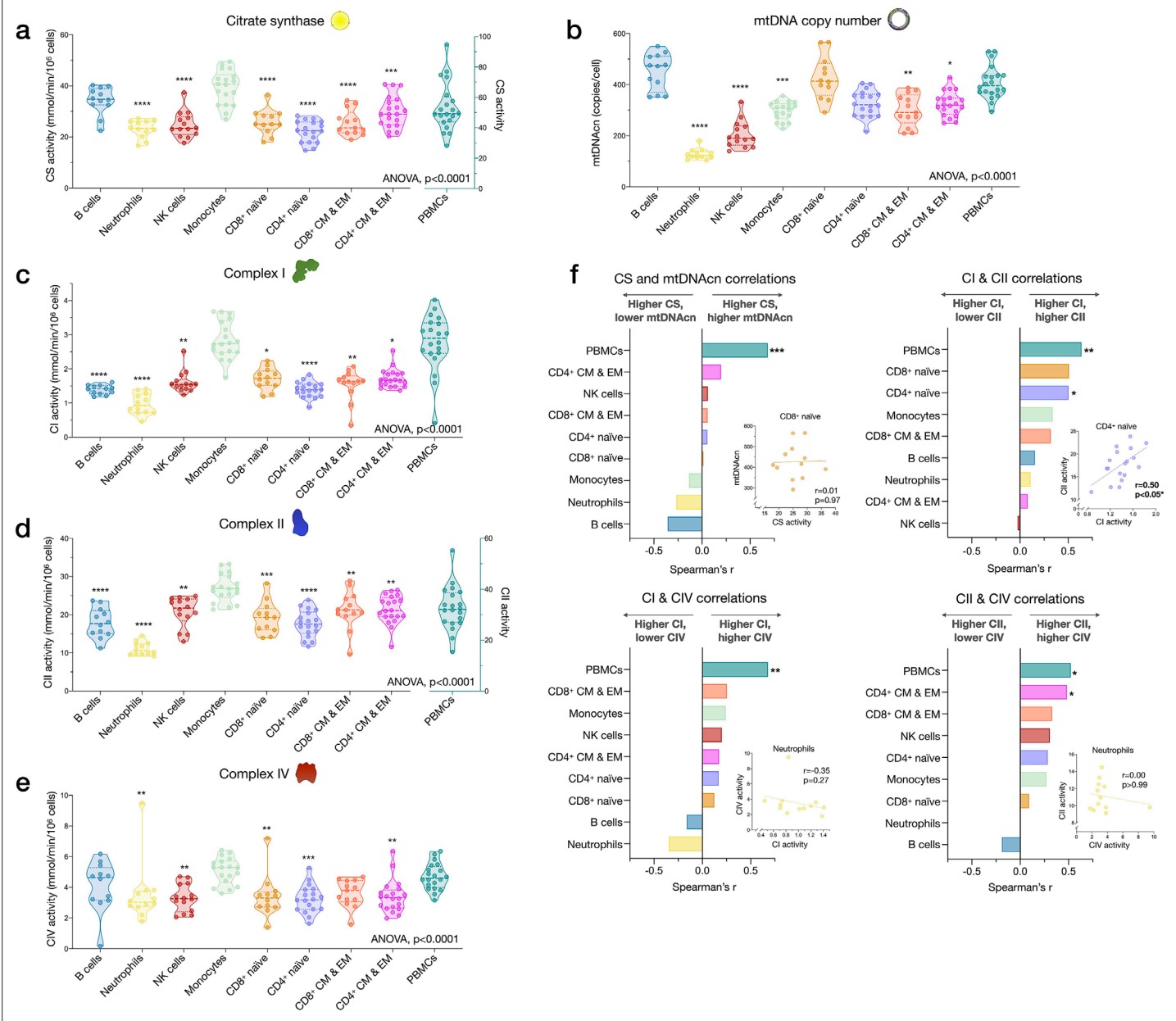

**Figure 4.** Cell subtype differences in mitochondrial content and RC function. (**a–e**) Violin plots illustrating immune cell type differences in mitochondrial features across cell subtypes and total PBMCs. For each individual, only the six most abundant cell types were analyzed (n=21 individuals, 12–18 per cell subtype). Dashed lines are median (thick) and 25th and 75th quartiles (thin). p-values from one-way nonparametric ANOVA Kruskal-Wallis test, post hoc Dunn's multiple comparisons relative to PBMCs. (**f**) Spearman's r inter-correlations of mitochondrial features across subtypes. Insets show the scatterplots for selected correlations. p<0.05*, p<0.01**, p<0.001***, p<0.0001****. PBMC, peripheral blood mononuclear cell; RC, respiratory chain.

The online version of this article includes the following figure supplement(s) for figure 4:

**Source data 1.** Cell subtype differences in mitochondrial content and RC function.

**Figure supplement 1.** Associations between subtype-specific enzymatic activities with mitochondrial features measured in PBMCs.

**Figure supplement 1—source data 1.** Associations between subtype-specific enzymatic activities with mitochondrial features measured in PBMCs.

women showed a trend (p=0.0047, Chi-square) for higher CS activity (range: 4–29%, g=0.20–1.52, *Figure 5a*) and higher CII activity than men (range: 1–10%, g=0.03–0.56, *Figure 5d*).

Other notable trends requiring validation in larger cohorts suggested that compared to women, men exhibited higher mtDNAcn in monocytes and neutrophils (range: 5–12%, g=0.37–0.73, *Figure 5b*), higher

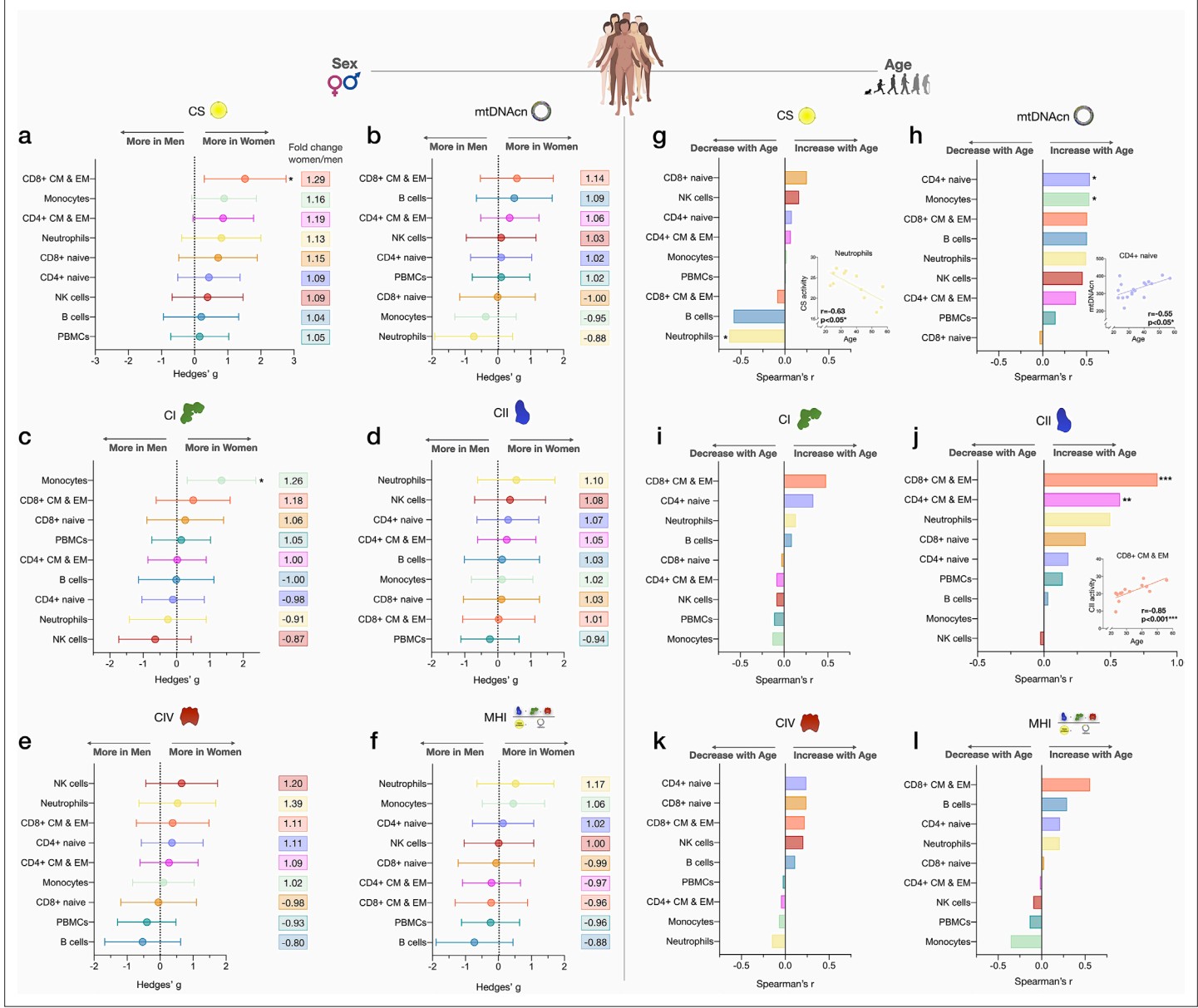

**Figure 5.** Associations of mitochondrial features with sex and age across cell subtypes. (**a–f**) Effect size of sex differences in mitochondrial activity across cell subtypes quantified by Hedges' g. The fold change computed from raw values is shown on the right. p-values from Mann-Whitney test, not adjusted for multiple comparisons. Error bars reflect the 95% CI on the effect size. (**g–l**) Association of age and mitochondrial features across cell subtypes. p-values from Spearman's r correlations, not adjusted for multiple comparisons. n=21 (11 women, 10 men), p<0.05*, p<0.01**, p<0.001***. CI, confidence interval.

The online version of this article includes the following figure supplement(s) for figure 5:

**Source data 1.** Associations of mitochondrial features with sex and age across cell subtypes.

CI activity in neutrophils and NK cells (range: 9–13%, g=0.26–0.64, *Figure 5c*), and higher CIV activity specifically in B cells (20%, g=0.53, *Figure 5e*). Cells exhibiting the largest degree of sexual dimorphism on the integrated MHI were neutrophils ( 17% higher in women, g=0.52) and B cells ( 12% higher in men, g=0.73, *Figure 5f*). In contrast, none of these potential differences were detectable in PBMCs, further illustrating the limitation of mixed cells to examine sex differences in mitochondrial function.

## Age associations with mitochondrial content and RC function

We then explored the association between mitochondrial features and age. With increasing age, CS activity was relatively unaffected except in neutrophils, where it significantly decreased by ~ 7% per

decade (r=−0.63, p<0.05, *Figure 5g*). In comparison, the correlation between age and mtDNAcn was positive among 7 out of 8 cell subtypes, with the exception being CD8$^+$ naïve T cells. Interestingly, CD8$^+$ naïve T cells are the cell type that exhibited the strongest age-related decline in abundance. CD4$^+$ naïve T cells and monocytes showed the largest age-related change in mtDNAcn, marked by a significant ~ 10% increase in the number of mitochondrial genome copies per cell per decade of life (r=0.54, p<0.05 for both, *Figure 5h*).

For RC function, an equal number of cell subtypes with either positive or negative correlations with age were found, except for CII (*Figure 5i–l*). Possibly due to the small sample size of our cohort, only CD4$^+$ and CD8$^+$ CM-EM T cells CII activity significantly increased with age (r=0.57 and 0.85, p<0.01 and 0.001, respectively, *Figure 5j*). However, CII activity tended to be positively correlated with age across all cell types except for monocytes and NK cells. In contrast, CI and CIV activities were only weakly associated with age, highlighting again differential regulation and partial 'biological independence' of different RC components. Of all cell types, CD8$^+$ CM-EM T cells showed the most consistent positive associations for all RC activities and age, most notably for CII where the enzymatic activity per cell increased a striking ~ 21% per decade (r=0.85, p<0.001).

Overall, these exploratory data suggest that age-related changes in CS activity, mtDNAcn, and RC function are largely cell-type specific. This conclusion is further reinforced by analyses of PBMCs where mitochondrial features consistently did not significantly correlate with age (rs=0.008–0.15, absolute values) (*Figure 5g–l*). Larger studies adequately powered to examine sex- and age-related associations are required to confirm and extend these results.

## Cell subtype distributions exhibit natural week-to-week variation

Samples collected weekly over 9 weeks from one repeat participant were used to (i) examine the stability of cell type-specific mitochondrial phenotypes described above, and (ii) quantify whether and how much mitochondrial content/function change over time (*Figure 6a*). First focusing on immune cell distribution, the cell subtype with the least week-to-week variation in abundance was CD8$^+$ EM (root mean square of successive differences [rMSSDs]=0.22, coefficient of variation [CV]=19.5%), which varied between 6.3% (week 2, highest) and 3.3% of all circulating cells (week 9, lowest) (*Figure 6— figure supplement 1*). Other subtypes such as CD4$^+$ TEMRA (min=0.02% to max=0.62%) and neutrophils (min=3.9% to max=31.8%) varied week-to-week by an order of magnitude (i.e., tenfold), similar to the between-person variation among the cohort (see *Figure 1e* and *Supplementary file 2*). The circulating abundance of B cells varied by up to 1.1-fold (min=0.86% to max=1.8%). Taken together, these time-course results illustrate the dynamic remodeling of circulating leukocyte populations (and therefore PBMC composition) within a single person.

The correlations between immune cell type composition at each week and PBMC mitochondrial features are shown in *Figure 6—figure supplement 2*. On weeks when the participant had higher circulating levels of EM and TEMRA CD4$^+$ and CD8$^+$ lymphocytes, most mitochondrial features were considerably lower in PBMCs. The associations between CBC-derived cell proportions and PBMCs mitochondrial features tended to be weaker and in opposite direction at the within-person level compared to the cohort (*Figure 6—figure supplement 2c-d*), but again document the influence of cell type composition on PBMC mitochondrial phenotypes.

## Mitochondrial content, mtDNAcn, and RC activity exhibit natural week-to-week variation

The six most abundant cell subtypes analyzed for this individual included: neutrophils, NK cells, monocytes, naïve and CM-EM subtypes of CD4$^+$ T cells, and naïve CD8$^+$ T cells. The robust cell type differences in mitochondrial content and RC activities reported above in our cohort were conserved in the repeat participant. This includes high mtDNAcn in CD8$^+$ naïve T cells (average across 9 weeks=400 copies/cell, 427 in the cohort) and lowest mtDNAcn in neutrophils (average=123 copies/cell, 128 in the cohort).

All mitochondrial metrics exhibited substantial weekly variation across the 9 time points. Different cell types showed week-to-week variation in CS, mtDNAcn, and RC activity ranging from 4.1% to 64.4% (*Figure 6b–f*). In most cases, the observed variation was significantly greater than the established technical variation in our assays (see *Supplementary file 3*), providing confidence that these

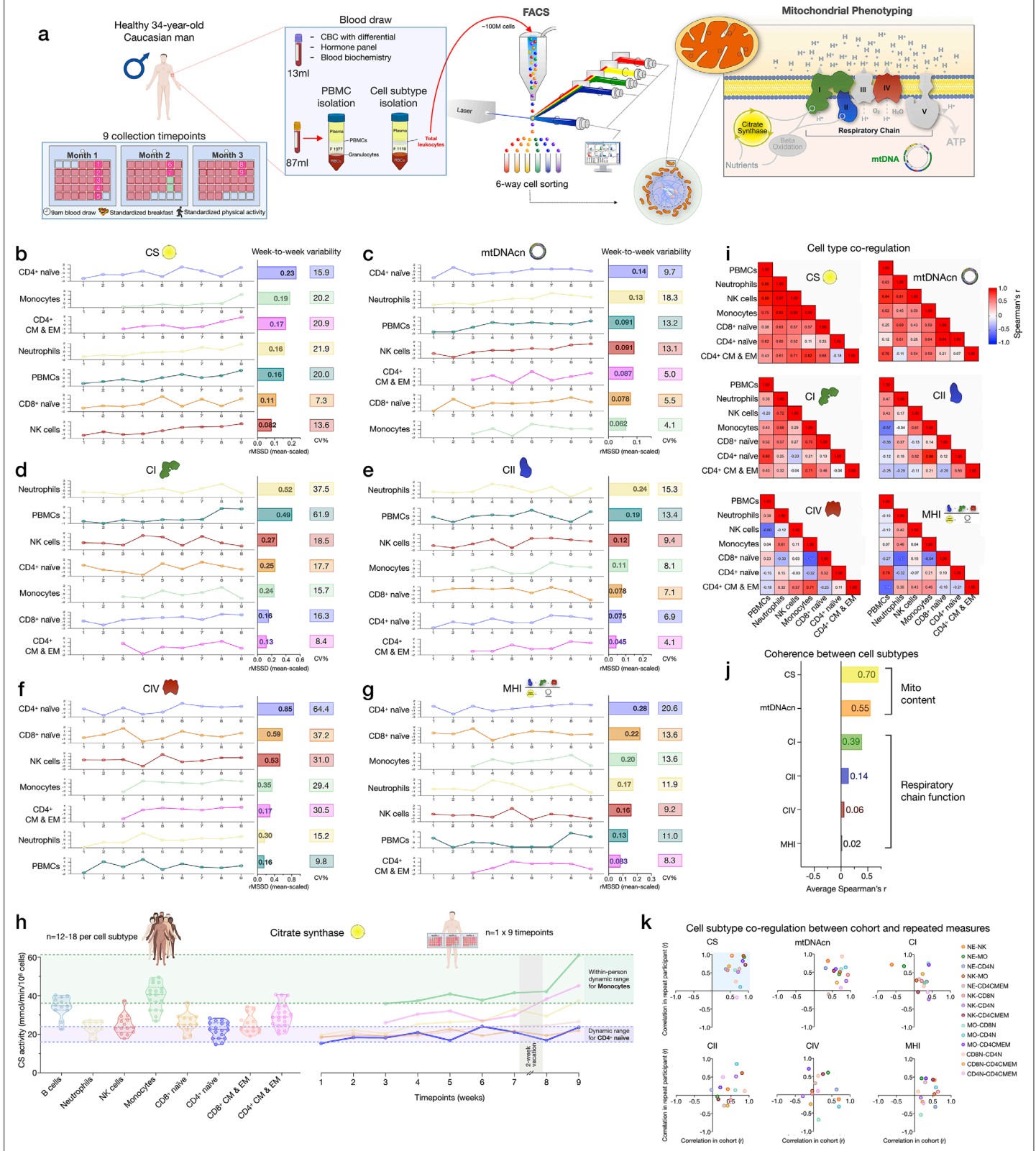

**Figure 6.** Within-person variability of mitochondrial features across cell subtypes. (**a**) Overview of the repeat participant design, including blood collection, processing, and analysis. Five million cells per purified cell subtype were used for analyses. All samples were collected, stored, and processed as a single batch with samples from the cohort. Mitochondrial phenotyping platform schematic adapted from **Figure 1a**, *Picard et al., 2018*. (**b–g**) Natural weekly variation for each mitochondrial feature across cell subtypes in the same person across 9 weeks represented as scaled centered data

*Figure 6 continued on next page*

*Figure 6 continued*

where one unit change represents a one standard deviation (SD) difference. Root mean square of the successive differences (rMSSDs) quantify the magnitude of variability between successive weeks. The coefficients of variation (CVs) quantify the magnitude of variability across all time points. Monocytes and CD4$^+$ CM-EM were not collected on weeks 1 and 2. (**h**) Side-by-side comparison of CS activity between the cohort (n=12–18 per cell subtype) and the repeat participant (n=7–9 time points) across cell subtypes. The dynamic range of two cell subtypes are highlighted: monocytes and CD4$^+$ naïve T cells. (**i**) Within-person correlation matrices between cell subtypes for each mitochondrial feature over 9 weeks, illustrating the magnitude of correlation (co-regulation) between cell subtypes. (**j**) Average inter-correlation across all cell subtypes by mitochondrial feature (calculated using Fisher z-transformation) indicating the degree of coherence within-person. (**k**) Comparison of co-regulation patterns among mitochondrial features between the cohort and the repeat participant. Each data point represents a cell subtype pair, indicating moderate agreement (data points in top right quadrant). CM, central memory; CS, citrate synthase; EM, effector memory.

The online version of this article includes the following figure supplement(s) for figure 6:

**Source data 1.** Within-person variability of mitochondrial features across cell subtypes.

**Figure supplement 1.** Within-person variability of cell subtype proportions over time.

**Figure supplement 1—source data 1.** Within-person variability of cell subtype proportions over time.

**Figure supplement 2.** Associations between subtype-specific and CBC cell proportions, and subtype-specific enzymatic activities with mitochondrial features measured in PBMCs.

**Figure supplement 2—source data 1.** Associations between subtype-specific and CBC cell proportions, and subtype-specific enzymatic activities with mitochondrial features measured in PBMCs.

**Figure supplement 3.** Variability of mitochondrial features across cell subtypes between the cohort and the repeat participant.

**Figure supplement 3—source data 1.** Variability of mitochondrial features across cell subtypes between the cohort and the repeat participant.

changes in mitochondrial content and function over time reflect real biological changes rather than technical variability.

We then asked how much the same metrics naturally vary within a person relative to differences observed between people in the heterogeneous cohort. Remarkably, the 9-week range of natural variation within the same person was similar to the between-person differences among the cohort. *Figure 6h* and *Figure 6—figure supplement 3* provide a side-by-side comparison of the cohort and repeat participant mitochondrial features (CS, mtDNAcn, RC activities, and MHI) on the same scale. A similar degree of variation (9.8–61.9%) was observed in PBMCs (*Figure 6b–g*), although again this variation may be driven in large part by variation in cell composition.

And as in the cohort, CS and mtDNAcn were also the most correlated across cell types (average $r_{z'}$=0.55–0.70) (*Figure 6i–k*, *Figure 6—figure supplement 2e*), indicating partial co-regulation of mitochondrial content features across cell subtypes.

## Mitotypes differ between immune cell subtypes

To examine cell type differences more fully, and in line with the concept of mitochondrial functional specialization, we performed exploratory analyses of cell subtype-specific mitochondrial phenotypes, or *mitotypes*, by mathematically combining multiple mitochondrial features in simple graphical representations (listed and defined in *Figure 7—figure supplement 1*). Each mitotype can be visualized as a scatterplot with two variables of interest as the x and y axes. Cell types that align diagonally from the origin of the plot indicate the same mitotype profile (*Figure 7a*).

The first mitotype examined mtDNA copies per cell (mtDNAcn) relative to mitochondrial content per cell (CS activity), creating a mitotype (mtDNAcn/CS) reflecting *mtDNA density per mitochondrion* (*Figure 7b*). Alone, the mtDNA density per mitochondrion mitotype provided remarkable separation of cell subtypes. Neutrophils and NK cells were low in both mtDNAcn and CS activity, B cells were high on both metrics, monocytes had the lowest mtDNA density, whereas CD8$^+$ naïve T cells exhibited the highest mtDNA density of all cell subtypes tested. *Figure 7c–e* illustrates other mitotypes including (i) CII activity per unit of mitochondrial genome (CII/mtDNAcn), as well as more complex combinations of variables such as (ii) CI activity per mtDNA (CI/mtDNAcn ratio on y-axis) in relation to mtDNA density (mtDNAcn/CS activity on x-axis), and (iii) CI activity per mitochondrial content (CI/CS, y-axis) in relation to mtDNA density (mtDNAcn/CS, x-axis). As *Figure 7b–e* shows, PBMCs generally exhibit a similar mitotype as innate immune cell subtypes (monocytes, NK cells, and neutrophils on the same diagonal), and are relatively distinct from lymphocyte subpopulations.

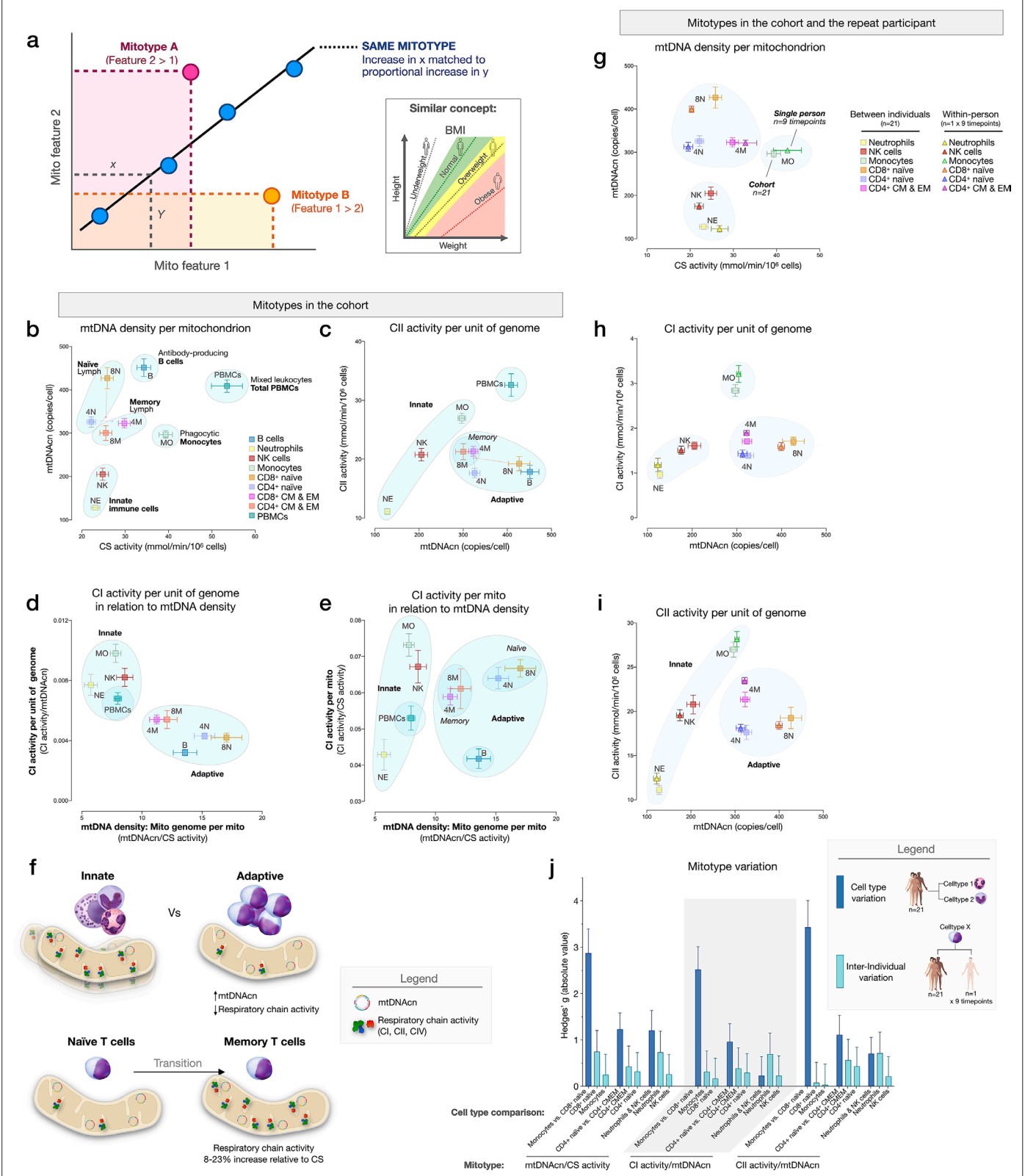

**Figure 7.** Mitotypes in purified leukocyte populations from the cohort and repeated measures. (**a**) Schematic illustrating the multivariate approach to generate and visualize mitotypes by putting into relation two or more mitochondrial features. Note the similarity and added insight relative to single metrics, similar to the integration of height and weight into the body mass index (BMI). (**b–e**) Selected mitotypes plotted for each cell subtype among the cohort. Data are means± SEM (n=12–18). Overlaid shaded areas denote general leukocyte categories, for visualization purposes only. (**f**) Summary

*Figure 7 continued*

of mitotype differences between (i) innate versus adaptive subdivisions, and (ii) naïve versus memory T cells. (**g–i**) Validation of subtype-specific mitotype differences in the repeat participant, illustrating the conserved nature of mitotypes across individuals. Only the six cell subtypes analyzed in the repeat participant are plotted. Data are means± SEM (n=7–9 for the repeat participant, 12–18 for the cohort). (**j**) Comparison of the magnitude of the difference (Hedges' g) in mitotypes between cell types, and between individuals. Dark blue bars indicate the magnitude of the dominant difference in mitotypes between cell subtypes. Light blue bars indicate the magnitude of the difference between the cohort and the repeat participant within a cell type. Error bars are the 95% CI of the effect size. CI, confidence interval.

The online version of this article includes the following figure supplement(s) for figure 7:

**Source data 1.** Mitotypes in purified leukocyte populations from the cohort and repeated measures.

**Figure supplement 1.** Operationalization and categorization of mitotypes.

This mitotype-based analysis revealed two main points. First, cells of the innate and adaptive immune subdivisions contain mitochondria that differ not only quantitatively in their individual metrics of mitochondrial content and RC activity, but also qualitatively, as illustrated by the distinct clustering of neutrophils, monocytes, and NK cells (innate immunity) within similar mitotype spaces, and the distinct clustering of all lymphocyte subtypes together in a different space. Compared to cells of the innate immune compartment, lymphocytes (adaptive immunity) had higher mtDNAcn and lower RC activity. Second, compared to naïve subsets of CD4[+] and CD8[+] T cells, which themselves have relatively distinct mitotypes (e.g., CII/mtDNAcn, *Figure 7c*), both memory CD4[+] and CD8[+] subtypes converged to similar mitotype spaces. Functionally, this naïve-to-memory transition is well known to involve a metabolic shift including changes in spare respiratory capacity, mitochondrial content, and glucose and fatty acid uptake (*Nicoli et al., 2018*; *van der Windt et al., 2012*). The mitotype analysis showed that compared to naïve cell subtypes, memory subtypes exhibit 26–29% lower mtDNA density per mitochondrion, but an 8–23% increase in RC activity per mitochondrion in CD4[+] T cells, although not in CD8[+] T cells (*Figure 7f*).

## Stability of mitotypes

The six cell subtypes analyzed in the repeat participant and the matching cell types for the cohort showed a high degree of agreement when plotted on the same mitotype plots. Again, cell types belonging to the innate and adaptive immune subdivisions clustered together, and naïve and memory subtype differences were similarly validated at the within-person level (*Figure 7g–i*), demonstrating the conserved nature of immune cell mitotypes in our sample. On average, the magnitude of variation between cell subtypes (e.g., monocytes vs. neutrophils) was 12.5-fold larger than the differences between the cohort and the repeat participant, indicating that immune cell subtypes have conserved mitotypes, exhibiting relative stability across individuals.

## Evidence for a sex- and age-related bias in mitotypes

We next sought to systematically examine if mitotypes differ between women and men. Mitotypes were organized into five categories of indices based upon their features, yielding a total of 16 mathematically distinct mitotypes (see *Figure 7—figure supplement 1*). For each mitotype, we quantified the magnitude of the difference between women and men by the effect size (g), ranked all mitotype × cell subtype combinations (16 mitotypes × 9 cell subtypes), and analyzed the distribution of these indices by sex. The majority of mitotypes reflecting mitochondrial RC activities per CS activity were higher in men (p<0.0001, Chi-square), while RC activity per mtDNA density (p<0.001) and RC activity per genome in relation to mtDNA density mitotypes (p<0.01) were predominantly higher in women (*Figure 8a*). The magnitude of sex differences ranged from 17% higher in men (CI/CS in CD4[+] CM-EM T cells, g=1.14) to 38% higher in women (CII/mtDNA density in neutrophils, g=1.37) (*Figure 8b*). The direction of sex differences for all mitotypes (e.g., higher in women or in men) with effect sizes is illustrated in *Figure 8c*. The average effect size across all mitotypes was 0.31 (small) in CD4[+] naïve T cells, compared to monocytes where the average effect size was 0.71 (medium). Compared to purified cell subtypes, the magnitude of sex differences in PBMCs was blunted.

Using the same approach, we then systematically quantified the relationship between mitotypes and age. Mitotypes reflecting RC activity per CS activity were predominantly positively correlated with age (p<0.05), while RC activities per genome in relation to mtDNA density were generally negatively correlated with age (p<0.05) (*Figure 8d*). This finding is consistent with the overall age-related

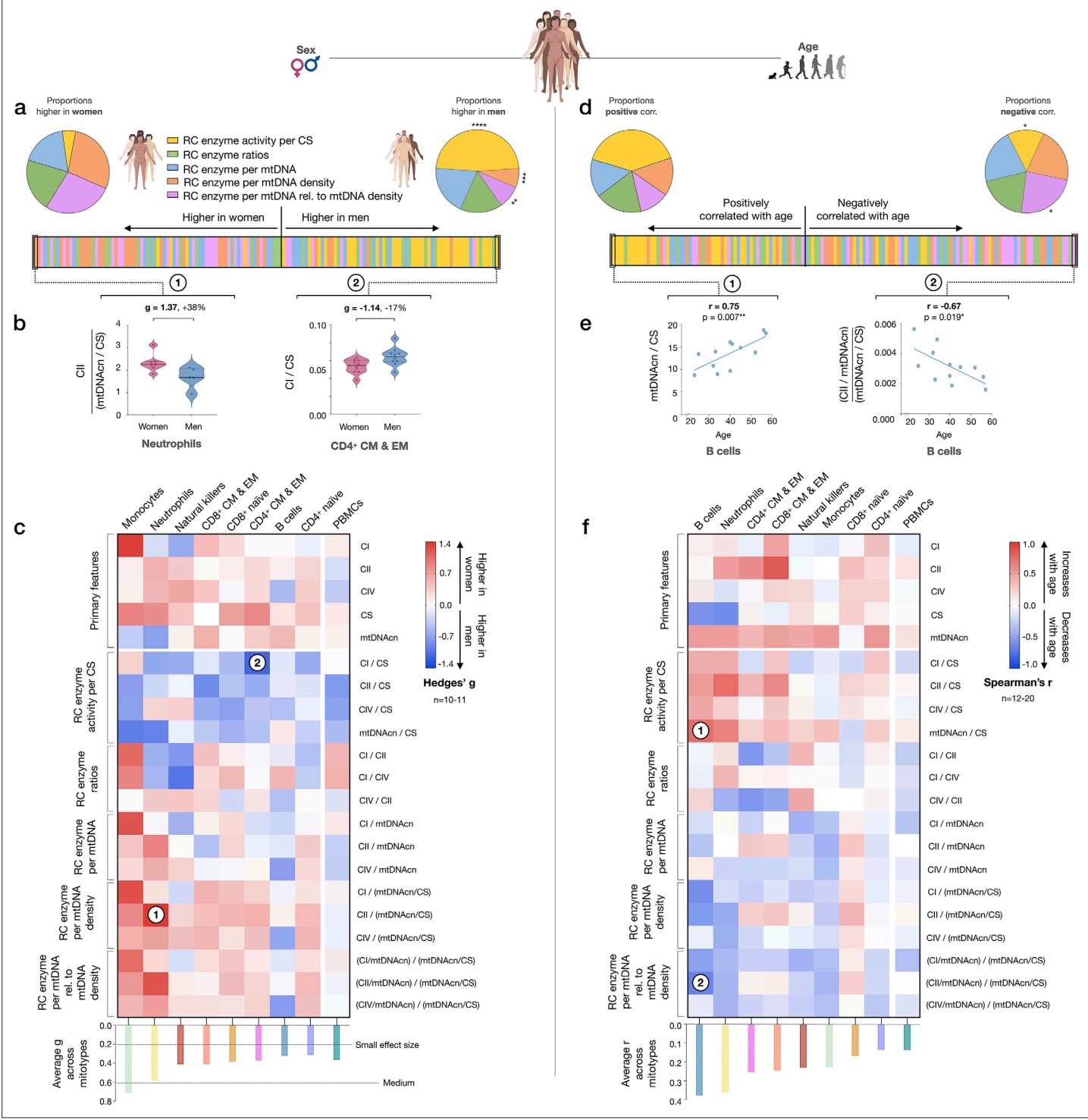

**Figure 8.** Mitotype distribution and strength of difference across sex and age. (**a**) Ranking of mitotype indices by the effect size (Hedges' g) between women and men. A total of 16 mitotype indices were computed, subdivided into five main color-coded categories (see **Figure 7—figure supplement 1**). Pie charts illustrate the proportion mitotypes belonging to each category that are either higher in women (*left*) or in men (*right*). p-values for enrichment of sexually dimorphic mitotypes are derived from the Chi-square test. (**b**) Violin plots illustrating the two mitotypes with the largest sex differences, both showing large effect sizes (g). (**c**) Heatmap of sex differences for primary measures of mitochondrial function (*top*) and multivariate mitotypes (*bottom*) across cell subtypes. The histogram at the bottom shows the average effect size across all mitotypes (calculated from absolute g values). (**d**) Ranking of mitotype indices by the strength and direction of their association with age, with enrichment analysis analyzed as for sex (Chi-square test). (**e**) Spearman's r correlations of mitotypes/cell type combinations with the strongest positive and negative associations with age. (**f**)

*Figure 8 continued on next page*

*Figure 8 continued*

Heatmap of the age correlations (Spearman's r) for primary features and composite mitotypes across cell subtypes. The histogram (*bottom*) shows the average effect size (r) for each cell subtype (calculated using absolute values and Fisher z-transformation). p<0.05*, p<0.01**, p<0.001***, p<0.0001****.

The online version of this article includes the following figure supplement(s) for figure 8:

**Source data 1.** Mitotype distribution and strength of difference across sex and age.

increase in mtDNAcn across cell subtypes, and could indicate a general decrease in the RC output per unit of mitochondrial genome with aging in immune cells. The strength of these correlations ranged from r=−0.67 to 0.75 (*Figure 8e*). The correlations of individual mitotypes with age for each cell subtype are shown in *Figure 8f*. Again, PBMCs showed among the weakest associations with either sex or age (*Figure 8c and f*). Thus, even if specific cell subtypes reveal consistent sex- and age-related differences, PBMCs offer modest to no sensitivity to detect these associations.

## Associations of blood biomarkers with subtype-specific mitochondrial features

Finally, to explore the source of inter-individual differences and within-person dynamics over time described above, we asked to what extent subtype-specific mitochondrial features were correlated with standard blood biomarkers, including a panel of sex hormones, inflammatory markers, metabolic markers, and standard clinical blood biochemistry (*Figure 9a*).

At the cohort level, sex- and age-adjusted partial correlations between blood biomarkers and cell subtype mitochondrial phenotypes were relatively weak (average absolute values $r_{z'}$=0.23, *Figure 9b*), indicating that circulating neuroendocrine, metabolic, and inflammatory factors are unlikely to explain a large fraction of the variance in inter-individual differences in mitochondrial biology. At the within-person level, week-to-week variation is independent of constitutional and genetic influences; additionally, behavior (e.g., levels of physical activity, sleep patterns, etc.) is more stable relative to between-person differences among a cohort of heterogenous individuals. Accordingly, compared to the cohort, the strength of biomarker-mitochondria associations was on average $r_{z'}$=0.39, or  70% larger in the repeat participant than the cohort (p<0.0001, comparing absolute correlation values) (*Figure 9c*). In particular, lipid levels including triglycerides, total cholesterol, and low- and high-density lipoproteins (LDL, HDL) were consistently positively correlated with markers of mitochondrial content (CS activity and mtDNAcn), with the largest effect sizes observed among innate immune cells: neutrophils, NK cells, and monocytes (*Figure 9c*, red area on the heatmap). In these cells, lipid levels accounted on average for  53% of the variance ($r^2$) in CS activity and  47% in mtDNAcn, possibly reflecting an effect of lipid signaling on mitochondrial biogenesis (*Iershov et al., 2019*; *Lindquist et al., 2018*; *Picard et al., 2012*; *Turner et al., 2007*).

Finally, we note that the existence of true, biologically meaningful interactions observed through repeated, within-person measures may be blunted or absent when observed cross-sectionally, at the group-level (*Fisher et al., 2018*). Accordingly, our data revealed major divergences in the correlation patterns between the cohort and the repeat participant (*Figure 9d–e*). Although limited by the small number of observations, these results highlight the value of repeated-measures study designs to examine the influence of metabolic and other humoral factors on human immune mitochondrial biology.

## Discussion

Developing approaches to quantify bioenergetic differences among various tissues and cell types is critical to define the role of mitochondria in human health and disease. Here, we isolated and phenotyped multiple molecularly defined immune cell subtypes and mixed PBMCs in >200 person-cell type combinations among a small diverse cohort of women and men, and in repeated weekly measures in the same participant. Our biochemical and molecular results confirm and extend previous knowledge of bioenergetic differences across human immune cell types established using extracellular flux analysis, protein and mtDNA quantification (*Chacko et al., 2013*; *Lee et al., 2021*; *Maianski et al., 2004*; *Pyle et al., 2010*). We also report preliminary evidence that mitochondrial phenotypes vary with age and sex, which PBMCs lack the sensitivity to detect. Importantly, our results confirm and quantify the

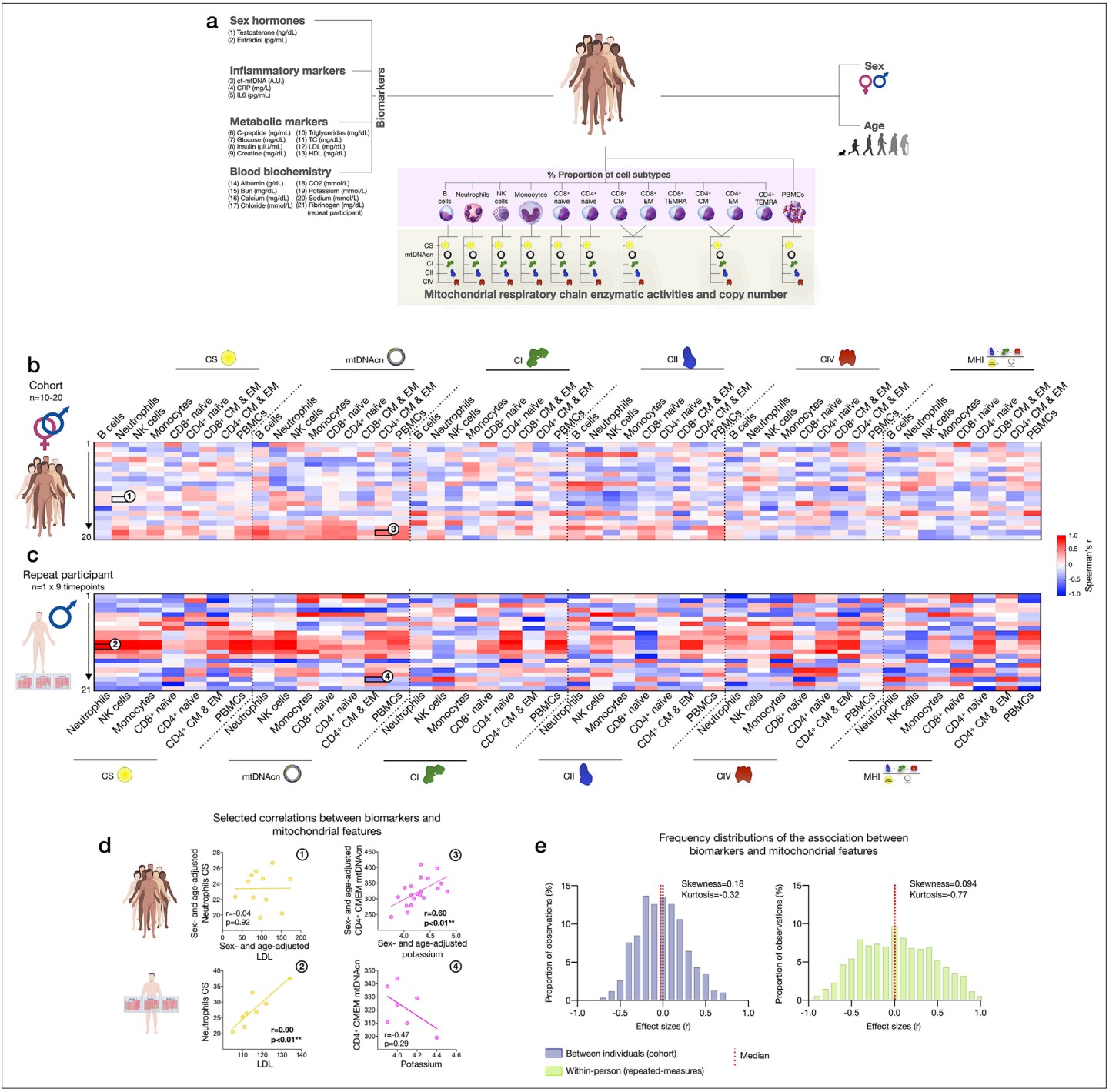

**Figure 9.** Association of blood biomarkers with mitochondrial parameters across cell subtypes and primary mitochondrial features. (**a**) Overview of blood biochemistry, hormonal, and metabolic biomarkers collected for each participant. (**b**) Sex- and age-adjusted correlations between blood biomarkers and mitochondrial features across cell subtypes for the cohort (n=10–20 per mito:biomarker combinations) are shown as a heatmap. (**c**) Same as (**b**), but using weekly measures of both mitochondrial features and biomarkers in the repeat participant. (**d**) Scatterplots of the indicated correlations between Neutrophils CS activity and LDL cholesterol (*left*), and CD4⁺ CM-EM mtDNAcn and potassium (K⁺) (*right*) for the cohort (*top row*) and the repeat participant (*bottom row*). (**e**) Frequency distributions of the aggregated effect sizes between biomarkers and mitochondrial features across cell subtypes for the cohort (total correlation pairs=1080) and the repeat participant (total correlation pairs=882). CM, central memory; CS, citrate synthase; EM, effector memory; LDL, low-density lipoprotein.

The online version of this article includes the following figure supplement(s) for figure 9:

**Source data 1.** Association of blood biomarkers with mitochondrial parameters across cell subtypes and primary mitochondrial features.

extent to which PBMCs mitochondrial measurements are confounded by (i) cell type composition, (ii) platelet contamination, (iii) mitochondrial properties across different cell subtypes, and (iv) by the dynamic remodeling of cell type composition and bioenergetics over time. In addition, large week-to-week, within-person variation in both cell subtype proportions and mitochondrial behavior points to heretofore underappreciated dynamic regulation of mitochondrial content and function over time in humans. Overall, our results provide standardized effect sizes of mitochondrial variation in relation to multiple key covariates, highlight the value of repeated-measures designs to carefully examine the mechanisms regulating mitochondrial health in humans, and call for the replication and extension of these findings in larger cohorts.

This study emphasizes the value of using purified cell populations over PBMCs for mitochondrial analyses. In many cases, associations with moderate to large effect sizes in specific cell subtypes were either not observed or blunted in PBMCs. For example, we found no correlation between age and PBMC MHI neither in this study nor in a previous one (*Karan et al., 2020*), whereas purified cell subtypes showed associations. Interestingly, in the mitotype plots, total PBMCs had similar mitotypes (along the same mitotype diagonal space) as cells of the innate subdivision, namely neutrophils, NK cells, and monocytes. If PBMCs were composed uniquely of a mixture of lymphocytes and monocytes, the natural expectation is that PBMCs would lie somewhere between the specific subsets that compose it. Instead, PBMCs occupy an entirely different and unexpected mitotype space. Additionally, our platelet depletion experiment leaves little doubt that platelet contamination skews the measurements of several mitochondrial features in PBMCs, with some features being apparently more affected than others, and yielding contradictory results: for example, PBMCs have higher CS activity values than any of the constituent cells (see *Figure 4a*). Although we cannot entirely rule out potential contamination of individual cell types with residual platelets, the FACS labeling, washing, and sorting procedures must produce the purest sample with the highest degree of biological specificity.

A major frontier for the human immunometabolism field consists in defining temporal trajectories of change in specific cell types (*Artyomov and Van den Bossche, 2020*). Achieving this goal promises to transform knowledge of immune and mitochondrial biology and allow for rational design of therapeutic approaches for immunometabolic conditions (*Artyomov and Van den Bossche, 2020*). A primary finding from our analyses is the natural within-person variation in mitochondrial features, providing initial insight into the temporal dynamics of immunometabolism in human leukocytes. Sorting immunologically defined cell subtypes removed the potential confound of week-to-week changes in cell type distributions, an inherent confounding variable in PBMCs, and therefore adds robustness to our observations. Mitochondrial features within immune cells exhibited state-like properties that varied by >20–30% week-to-week, warranting future, adequately-powered studies of causal influences. Previously, in PBMCs, up to 12% of the inter-individual variation in MHI was found to be attributable to positive mood (e.g., excited, hopeful, inspired, and love) the night prior to blood draw (*Picard et al., 2018*), implying that psychosocial factors could in part contribute to dynamic variation in leukocyte mitochondrial function over 12–48 hr. However, limitations in this prior study, including the use of PBMCs and a single measurement time point for MHI, call for additional studies to disentangle the independent contributions of behavioral, psychosocial, nutritional, and other factors on specific mitochondrial features. Importantly, mitochondrial changes in the present study took place within less than 1 week. Tracking dynamic changes in biological markers with high intra-individual variability, such as cortisol (*Segerstrom et al., 2017*), necessarily requires repeated-measures designs with sufficient temporal resolution. Therefore, the present results and others show that establishing the exact temporal dynamics of leukocyte mitochondrial variations, and immunometabolism in general, will require repeated assessments with substantially greater temporal resolution than weekly measures.

Animal studies have consistently identified sexually dimorphic mitochondrial features, such as greater mitochondrial content in females (reviewed in *Ventura-Clapier et al., 2019*). Likewise, in humans, PBMCs from women had greater CS activity and greater CI and CII-mediated respiration (*Silaidos et al., 2018*). Our data show similar changes in enzymatic activities for most, but not all, cell types, suggesting that the magnitudes of sex differences are likely cell-type specific. Therefore, methods offering a sufficient level of biological specificity deployed in adequately powered samples are needed to reproducibly and accurately quantify sex differences among immune cell mitochondria. We also note that the binary definition of sex used in this study (i.e., sex assigned at birth) paints a

rather incomplete picture. Exploring the interplay between mitochondria, sex and gender in humans will require more refined approaches that consider a range of biological characteristics (e.g., hormones, chromosomes, and anatomy), alongside gender identity and its proceeding gendered exposures (e.g., differential diet or involvement in physical activity depending on gender/sex) (*Fausto-Sterling, 2005*; *Johnson and Repta, 2012*; *Ritz et al., 2014*).

In relation to age, age-related decline in mtDNAcn has consistently been reported in whole blood (*Mengel-From et al., 2014*; *Verhoeven et al., 2018*), PBMCs (*Zhang et al., 2017*), and skeletal muscle tissue (*Hebert et al., 2015*; *Short et al., 2005*), although not in liver (*Wachsmuth et al., 2016*). However, the interpretation of these results must take into account the existence of cell mixtures and platelet contamination, particularly for blood and PMBCs (*Banas et al., 2004*; *Urata et al., 2008*). In one large adult cohort study, accounting for cell type distribution and platelet count through measurement and statistical adjustments eliminated initial associations between mtDNAcn and age (*Moore et al., 2018*), suggesting that the *apparent* age-related decline in mtDNAcn in human blood in fact reflects a change in blood composition (fewer platelets in older people, explaining why mtDNAcn appears lower). In purified immune cell subtypes from our small cohort, we observed the opposite association: an age-related positive correlation with mtDNAcn. Whereas this finding was unexpected based on prior whole blood and tissue studies, it could be explained by the well-known cellular response qualitative changes in mtDNA or other bioenergetic alterations (*Picard, 2021*). Mutations and deletions in mtDNA accumulate with age (*Ye et al., 2014*; *Zhang et al., 2017*), and mtDNA defects can trigger the compensatory upregulation of mtDNAcn to counteract that loss of intact mitochondria (*Giordano et al., 2014*; *Yu-Wai-Man et al., 2010*). Therefore, we speculate that the observed positive correlation of cell type-specific mtDNAcn with age in our sample could reflect compensatory upregulation of mtDNA replication. Alternatively, this correlation could reflect impaired autophagic removal of mitochondria in aging cells, consistent with recent results suggesting that CD4[+] T cells have impaired clearance of dysfunctional mitochondria (*Bektas et al., 2019*). Interestingly, the only cell type examined that did not exhibit positive correlation between mtDNAcn and age was CD8[+] naïve T cells, which is also the only cell type whose abundance in circulation significantly declines with advancing age. The basis for the direction of this association requires further investigation.

Some limitations of this study must be noted. Although this represents, to our knowledge, the largest available study of mitochondrial biochemistry and qPCR in hundreds of human samples, the sample size of the cohort was small and the power to examine between-person associations was limited. Women and men were equally represented, but the sample size precluded stratification of all analyses by sex. Sex and age analyses are exploratory and findings need to be validated by future adequately powered studies. Likewise, the exhaustive repeated-measures design was carried out in only one participant and should be regarded as proof-of-concept. Additionally, because our mitochondrial phenotyping platform required ~$5 \times 10^6$ cells per sample, we could only collect the six most abundant cell subtypes from each participant, which in some instances reduced the final sample size for different cell subtypes. In order to accommodate the minimum cell number per sample, CM and EM subtypes were pooled, although they may exhibit differences that are not resolved here. Furthermore, we recognize that additional cell surface markers may be useful to identify other cell populations (e.g., activated or regulatory lymphocyte subtypes). Finally, we did not test participants for cytomegalovirus status, which could influence the proportion of immune cell subtypes.

Furthermore, our analysis focused on RC activity, which performs electron transport and generates the mitochondrial membrane potential ($\Delta\Psi m$) across the inner mitochondrial membrane (*Nicholls and Ferguson, 2013*). Besides being used for ATP synthesis by complex V, RC activity and membrane potential also contributes to reactive oxygen species production, calcium handling, and gene expression regulation, among other cellular processes (*Brand et al., 1976*; *Hernansanz-Agustín and Enríquez, 2021*; *Martínez-Reyes et al., 2016*; *Picard et al., 2014*). Thus, the observed cell type differences in mitochondrial content or RC activities across human immune cell subtypes likely reflect not only cellular ATP demand, but also the unique immunometabolic, catabolic/anabolic, and signaling requirements among different immune cell subtypes that contribute to produce cell-specific mitotypes.

Overall, our study of mitochondrial profiling in circulating human immune cells filled three main knowledge gaps. First, it quantified confounds for PBMCs and showed how PBMCs fail to capture age- and sex-related mitochondrial recalibrations in specific immune cell populations, which is

important for the design of future studies. Second, mitochondrial profiling precisely documented large-scale, quantitative differences in CS activity, mtDNAcn, and RC enzyme activities between well-known immune cell subtypes, contributing to our knowledge of the distinct metabolic characteristics among circulating immune cell types in humans. The qualitative and quantitative divergences were particularly emphasized by mitotypes, which highlighted conserved multivariate phenotypic features between lymphoid- and myeloid-derived immune cells, and between naïve and memory lymphocyte states. Third, this study documents potentially large week-to-week variation of mitochondrial activities. This has important implications for discovering individual-level processes (*Fisher et al., 2018*) that may dynamically influence mitochondrial biology, and should be further examined in future studies. Taken together, this work provides foundational knowledge and a resource to develop interpretable blood-based assays of mitochondrial health.

## Materials and methods

### Participants and procedures

A detailed account of all methods and procedures is available in Appendix 1. The study was approved by New York State Psychiatric Institute (Protocol #7618) and all participants provided written informed consent for the study procedures and reporting of results. Healthy adults between 20 and 60 years were eligible for inclusion. Exclusion criteria included severe cognitive deficit, symptoms of flu or other seasonal infection 4 weeks preceding the visit, involvement in other clinical trials, malignancy or other clinical condition, and diagnosis of mitochondrial disease. The main study cohort included 21 individuals (11 women, 10 men), with a mean age of 36 years (SD=11, range: 23–57); there were 2 African Americans, 7 Asians, and 12 Caucasians. Morning blood samples (100 ml) were drawn between 9 and 10 a.m. from the antecubital vein and included one EDTA tube for CBC, two SST coated tubes for hormonal measures and blood biochemistry, and 11 Acid Dextrose A (ACD-A) tubes for leukocyte isolation and mitochondrial analyses. See *Figure 1a–b* for an overview of participants and procedures.

Additionally, repeated weekly measures were collected across 9 weeks from one healthy Caucasian man (author M.P., 34 years old) to assess within-person variability in mitochondrial measures and immune cell type distribution. To standardize and minimize the influence of nutritional and behavioral factors in the repeat participant, repeated measures were collected at the same time (9:00 a.m.), on the same day of the week (Friday), after a standardized breakfast (chocolate crepe), ~30–60 min after a regular bicycle commute to study site.

### PBMCs and leukocyte isolation

A detailed version of the materials and methods is available in Appendix 1 of this article. Briefly, PBMCs were isolated on low-density Ficoll 1077, and total leukocytes were separated on Ficoll 1119, centrifuged at 700×$g$ for 30 min at room temperature. Leukocytes were collected, washed, and centrifuged twice at 700×$g$ for 10 min to reduce platelet contamination. Pellets of 5×10$^6$ PBMCs were aliquoted and frozen at – 80°C for mitochondrial assays.

### Immunolabeling and fluorescence-activated cell sorting

Antibody cocktails for cell counting (Cocktail 1) and cell sorting (Cocktail 2) were prepared for FACS. The following cell subtypes were identified: neutrophils, B cells, monocytes, NK cells, naïve CD4$^+$ and CD8$^+$, CM CD4$^+$ and CD8$^+$, EM CD4$^+$ and CD8$^+$, and terminally differentiated EM cells re-expressing CD45RA (TEMRA) CD4$^+$ and CD8$^+$ (see *Figure 1—figure supplement 1* and *Supplementary file 1* for overview, and *Supplementary file 4* for cell surface markers and fluorophores). A 2×10$^6$ cell aliquot was labeled with Cocktail 1. The remainder of total leukocytes (~100×10$^6$ cells) were incubated with Cocktail 2, washed, and used for FACS at final concentration of 20×10$^6$ cells/ml.

Leukocytes were sorted using a BD Influx cell sorter using a 100-µm size nozzle. Sorting speed was kept around 11,000–12,000 events/s. Cell concentration for sorting was measured at about 15×10$^6$ cells per ml. For each participant, 1×10$^6$ cells (Cocktail 1 panel) were run first to calculate the potential yield of each subpopulation (total cell number × percentage of each subpopulation). The variable proportions of cell subtypes from person-to-person (provided in full in *Supplementary file 2*) determined which cell subtypes were collected from each participant, and 5×10$^6$ cell aliquots of the six most abundant subpopulations were sorted. Purity checks were performed on all sorted

subpopulations to ensure the instrument performance was good enough to reach the sorted population purity >95%. Data were processed using FCS Express 7 software (see *Figure 1—figure supplement 2* for gating strategy).

## Mitochondrial enzymatic activities

Sorted cell subtypes were centrifuged at 2000×*g* for 2 min at 4°C and stored in liquid nitrogen (− 170°C) for 4–12 months until mitochondrial biochemistry and mtDNAcn analyses were performed as a single batch. Cell pellets ($5×10^6$) were mechanically homogenized with a tungsten bead in 500 µl of homogenization buffer as previously described (*Picard et al., 2018*) (see Appendix 1 for details).

Mitochondrial enzyme activities were quantified spectrophotometrically for CS, cytochrome c oxidase (COX, Complex IV), succinate dehydrogenase (SDH, Complex II), and NADH dehydrogenase (Complex I) as described previously (*Picard et al., 2018*) with minor modifications described in Appendix 1. Each sample was measured in triplicates. Mitochondrial enzymatic activities were measured on a total of 340 samples (including 136 biological replicates), for a total of 204 unique participant-cell combinations. The technical variation for each enzyme, for each cell type, is detailed in *Supplementary file 3*.

mtDNAcn was determined as described previously (*Picard et al., 2018*) using two different Taqman multiplex assays for ND1 (mtDNA) and B2M (nDNA), and for COX1 (mtDNA) and RnaseP (nDNA). mtDNAcn was calculated from the ΔCt obtained by subtracting the mtDNA Ct from nDNA Ct for each pair ND1/B2M and COX1/RNaseP, and mtDNAcn from both assays was averaged to obtain the final mtDNAcn value for each sample. The CVs for mtDNA for each cell subtype are detailed in *Supplementary file 3* (average 5.1%).

## Platelet depletion in PBMCs

A further nine community-dwelling older adults (mean age=79, range: 64–89, 4 women and 5 men, including 7 White and 2 African American participants) were recruited for active platelet depletion experiments. Exclusion criteria included diseases or disorders affecting the immune system including autoimmune diseases, cancers, immunosuppressive disorders, or chronic, severe infections; chemotherapy or radiation treatment in the 5 years prior to enrollment; unwillingness to undergo venipuncture; immunomodulatory medications including opioids and steroids; or more than two of the following classes of medications: psychotropics, anti-hypertensives, hormones replacement, or thyroid supplements. Participants were recruited from a volunteer subject pool maintained by the University of Kentucky Sanders-Brown Center on Aging. The study was conducted with the approval of the University of Kentucky Institutional Review Board. Morning blood samples (20 ml) were collected by venipuncture into heparinized tubes. PBMCs were isolated from diluted blood by density gradient centrifugation 800×*g* for 20 min using Histopaque. Buffy coats were washed once, and cells were counted using a hemocytometer. PBMCs (20–30 M) were cryopreserved in liquid nitrogen in RPMI-1640 with 10% fetal bovine serum (FBS) and 10% dimethyl sulfoxide (DMSO) until further processing.

For active platelet depletion, PBMCs were first thawed at room temperature and centrifuged at 500×*g* for 10 min, counted and divided into two aliquots, including $10×10^6$ cells for total PBMCs, and $11×10^6$ cells for platelet depletion. The total PBMCs were centrifuged at 2000×*g* for 2 min at 4°C and frozen as a dry pellet at − 80°C until processing for enzymatic assays and qPCR. The PBMCs destined for platelet depletion were processed immediately and depleted of platelets following the manufacturer's procedures using magnetically coupled antibodies against the platelet marker CD61. This experiment yielded three samples per participant, including total PBMCs, platelet depleted PBMCs, and enriched platelets. Each sample was processed in parallel for RC enzymatic activity assays and mtDNAcn as described above.

## Statistical analyses

To adjust for potential order and batch effects across the 340 samples (31 samples per 96-well plate, 17 plates total), a linear adjustment was applied to raw enzymatic activity measures to ensure consistency between the first and last samples assayed. Samples from both the cohort and the repeat participant were processed and analyzed as a single batch, ensuring directly comparable data.

Throughout, standardized effect sizes between cell subtypes and between groups were computed as Hedge's g (g). Mann-Whitney t-tests were used to compare sex differences in cell type proportions

and mitochondrial measures. Spearman's r (r) was used to assess the strength of associations between continuous variables such as age and circulating proportions of cell subtypes. To assess to what extent mitochondrial features are correlated across cell subtypes (co-regulation) and to calculate the average correlation across mitotypes, Spearman's r matrixes were first computed and transformed to Fisher's z', and then averaged before transforming back to Spearman's r ($r_{z'}$). One-way nonparametric ANOVA Kruskal-Wallis with post hoc Dunn's multiple comparison tests were used to compare cell type mitochondrial measures in different cell subtypes and PBMCs. Between- and within-person variation were characterized using CVs. The rMSSD was computed to quantify the magnitude of variability between successive weeks for repeated measures. Chi-square tests were computed to compare proportion of mitotype indices categories (enzyme activity per CS, enzyme ratios, enzyme per mtDNA, enzyme per mtDNA density, and enzyme per mtDNA relative to mtDNA density) by age (lower vs. higher with increased age) and sex (lower vs. higher in men). Finally, one-way nonparametric Friedman tests with post hoc Dunn's multiple comparisons were used to compare mitochondrial measures in platelet-depleted PBMCs, enriched platelets PBMCs, and total PBMCs. Statistical analyses were performed with Prism 8 (GraphPad, CA), R version 4.0.2, and RStudio version 1.3.1056. Statistical significance was set at $p < 0.05$.

## Acknowledgements

Work of the authors is supported by the Wharton Fund and NIH grants MH119336, GM119793, MH122706, AG066828, AG056635, AG026307, and UL1TR001873. These studies used the resources of the Irving Cancer Center Core Facility funded in part through center grant P30CA013696.

## Additional information

### Funding

| Funder | Grant reference number | Author |
|---|---|---|
| Nathaniel Wharton Fund | | Martin Picard |
| National Institutes of Health | MH119336 | Martin Picard |
| National Institutes of Health | P30CA013696 | Wei Wang |
| National Institutes of Health | GM119793 | Martin Picard |
| National Institutes of Health | MH122706 | Martin Picard |
| National Institutes of Health | AG066828 | Martin Picard |
| National Institutes of Health | AG056635 | Rebecca G Reed |
| National Institutes of Health | AG026307 | Suzanne C Segerstrom |
| National Institutes of Health | UL1TR001873 | Martin Picard |

The funders had no role in study design, data collection and interpretation, or the decision to submit the work for publication.

### Author contributions

Shannon Rausser, Data curation, Formal analysis, Investigation, Visualization, Writing – original draft, Writing – review and editing; Caroline Trumpff, Conceptualization, Data curation, Formal analysis, Visualization, Writing – review and editing; Marlon A McGill, Methodology, Writing – review and editing; Alex Junker, Formal analysis, Visualization, Writing – review and editing; Wei Wang, Methodology,

Resources, Writing – review and editing; Siu-Hong Ho, Conceptualization, Methodology, Resources, Writing – review and editing; Anika Mitchell, Investigation, Writing – review and editing; Kalpita R Karan, Data curation, Methodology, Writing – review and editing; Catherine Monk, Writing – review and editing; Suzanne C Segerstrom, Formal analysis, Writing – review and editing; Rebecca G Reed, Conceptualization, Formal analysis, Writing – review and editing; Martin Picard, Conceptualization, Funding acquisition, Project administration, Supervision, Writing – original draft, Writing – review and editing

### Author ORCIDs
Shannon Rausser ⓘD http://orcid.org/0000-0003-0425-1571
Wei Wang ⓘD http://orcid.org/0000-0001-9890-2122
Martin Picard ⓘD http://orcid.org/0000-0003-2835-0478

### Ethics
Human subjects: The study was approved by New York State Psychiatric Institute (Protocol #7618) and all participants provided written informed consent for the study procedures and reporting of results.

### Decision letter and Author response
Decision letter https://doi.org/10.7554/eLife.70899.sa1
Author response https://doi.org/10.7554/eLife.70899.sa2

---

## Additional files

### Supplementary files
• Appendix 2—figure 1—source data 1. Mitochondrial health index and coherence of mitochondrial features across cell subtypes.

• Supplementary file 1. Leukocyte subtypes included in the study. Immune cell subtypes included in this study, including a brief summary of their functions and cell surface markers used for immunolabeling and FACS. FACS, fluorescence-activated cell sorting.

• Supplementary file 2. CBC- and FACS-based cell proportions for all study participants and time points. Participants are ordered by age. CBC measurements were performed using a Sysmex XN-9000 instrument, and FACS-based cell proportions were determined using a BD Influx cell sorter (see Appendix 1 for details). CBC, complete blood count; FACS, fluorescence-activated cell sorting.

• Supplementary file 3. Technical variation for each mitochondrial assay and calculated MHI by cell type. Coefficients of variation (CVs) across 2–5 biological replicates (different 5 M cell pellet isolated from the same blood draw) for each cell subtype and PBMCs (see Appendix 1 for details). MHI, mitochondrial health index; PBMC, peripheral blood mononuclear cell.

• Supplementary file 4. Recipes for antibody cocktails used to detect cell surface markers for FACS-based cell proportions and sorting (see Appendix 1 for details).
 FACS, fluorescence-activated cell sorting.

• Transparent reporting form

• Source data 1. Technical variation for each mitochondrial assay and calculated MHI by cell type.

### Data availability
All data generated and analyzed during this study, including mitochondrial biochemistry, mtDNA content, and blood chemistry, cell counts from CBC and flow cytometry, and de-identified participant information are included in the supporting data files. Source data files have been provided for Figures 1–9, and for figure supplements (Figure 1—figure supplement 3, Figure 2—figure supplement 1, Figure 4—figure supplement 1, Figure 4—figure supplement 2, Figure 6—figure supplement 1, Figure 6—figure supplement 2, Figure 6—figure supplement 3, and Supplementary file 3). Requests for resources or other information should be directed to and will be fulfilled by the corresponding author.

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

# Appendix 1

## Supplemental methods and procedures

This Appendix is an extended version of the Materials and methods section in the main text. It includes additional technical details to facilitate the implementation and reproducibility of methods used in this study.

### Blood collection

A total of 100 ml of blood was drawn from the antecubital vein for each participant and included one EDTA tube (Cat# BD367841, 2 ml) for CBC, two SST coated tubes (Cat# BD367986, 5 ml) for hormonal measures and blood biochemistry in serum, and 11 Acid Dextrose A (ACD-A) tubes (Cat# BD364606, 8.5 ml) for leukocyte isolation and mitochondrial analyses, in order of collection. All tubes were gently inverted 10–12 times immediately after draw to ensure proper mixing. The EDTA and SST tubes for hematological and blood biochemistry were processed by the Center for Advanced Laboratory Medicine (CALM) Lab, and the ACD-A tubes were further processed for cell isolation.

### PBMCs and leukocyte isolation

Ficoll 1077 and 1119 (Sigma-Aldrich), Hanks' Balanced Salt Sodium (HBSS) without phenol red, calcium and magnesium (Life Technologies, Cat# 14175103) supplemented with 2 % bovine serum albumin (BSA) (Sigma-Aldrich, Cat# A9576) (HBSS/BSA), HBSS/BSA supplemented with 1 mM of EDTA (Sigma-Aldrich, Cat# E9884) (HBSS/BSA/EDTA), and FBS (Thermo Fisher Scientific, Cat# 10437036) were brought to room temperature overnight. PBMCs were isolated on 15 ml of low-density Ficoll 1077 in a 50-ml conical tube, and total leukocytes were separated on 15 ml of higher density Ficoll 1119 distributed across seven conical tubes. Blood was first pooled and diluted with HBSS/BSA in a 1:1 ratio, and 25 ml of diluted blood was carefully layered on Ficoll and then centrifuged immediately at 700×$g$ for 30 min (no brake) in a horizontal rotor (swing-out head) tabletop centrifuge, at room temperature. Immediately after centrifugation, cells at the interface were collected and washed in 50 ml HBSS/BSA and centrifuged at 700×$g$ for 10 min. Supernatants were discarded and leukocyte pellets were washed again in HBSS/BSA/EDTA and centrifuged at 700×$g$ for 10 min to limit platelet contamination. Low concentration EDTA (1 mM) was used to prevent cell-cell adhesion or platelet activation, but a higher concentration was not used to avoid perturbing downstream mitochondrial assays.

To perform cell count, both (i) PBMCs (1:10 dilution) and (ii) total leukocytes (1:100 dilution) were resuspended in 1 ml of HBSS/BSA/EDTA and counted on the Countess II FL Automated Cell Counter (Thermo Fisher Scientific, Cat# AMQAF1000) in a 1:1 dilution with trypan blue. Counts were performed in duplicates. If the difference between duplicates >10%, the count was repeated and the grand mean of the cell counts taken. Pellets of 5 million PBMCs were aliquoted and frozen at – 80°C for mitochondrial assays.

### Immunolabeling of cell surface markers

Two antibody cocktails, meant for (i) cell counting (Cocktail 1) and (ii) cell sorting (Cocktail 2), were prepared for FACS (see *Supplementary file 4* for details). Antibodies were gently mixed, kept at 4°C, and shielded from light to avoid bleaching. Cocktail 1 (containing cell surface markers for activated T lymphocytes) was prepared with 17.5 µl HBSS/BSA/EDTA and 2.5 µl per antibody (13 markers, 32.5 µl total antibody mix), for a total of 50 µl. Cocktail 2 was prepared with 200 µl HBSS/BSA/EDTA and 25 µl per antibody (12 markers, 300 µl total sorting antibody mix), for a total of 500 µl.

Prior to each study visit, cell collection tubes (Cat# 352063, polypropylene, 5 ml) were coated with 4.5 ml of DMEM/ 10% FBS media to minimize cell-tube adhesion and maximize the recovery of sorted cells. Tubes were incubated for 24 hr at room temperature and stored at 4°C until use, and decanted prior to use. Two coated polypropylene tubes were used for the FACS-ready antibody-labeled leukocytes, and an additional 60 coated polypropylene falcon tubes were decanted and 500 µl of media (DMEM/ 10% FBS) was added to receive sorted cells.

Prior to immunolabeling, total leukocytes were incubated with blocking buffer (to block non-specific binding to FC receptors) at a 1:10 dilution and incubated at room temperature for 10 min. A 2 million cell aliquot was diluted to a final volume of 100 µl with HBSS/BSA/EDTA and combined

with 50 µl of Cocktail 1. The remainder of total leukocytes (~100 M cells) were incubated with 500 µl of Cocktail 2 for 20 min in the dark, at room temperature. Both cell preparations were then washed with 5 ml of HBSS/BSA/EDTA and centrifuged at 700×$g$ for 5 min. Using the propylene tubes, Cocktail 1 cells were resuspended in 200 µl of HBSS/BSA/EDTA, and total leukocytes for FACS were resuspended to final concentration of 20 million cells/ml with HBSS/BSA/EDTA.

## Fluorescence-activated cell sorting

Cells labeled with the Cocktail 1 (counting) panel was only used for data acquisition and phenotype analysis. Cells labeled with the Cocktail 2 (sorting) panel was FACS sorted using a BD Influx cell sorter to isolate the subpopulations from peripheral blood. The sorter was controlled using BD FACS Sortware. Cells were sorted using 100 µm size nozzle and under the sheath pressure of 20 psi. Sorting speed was kept around 11,000–12,000 events/s. Cell concentration for sorting was measured at about 15×10⁶ cells per ml. Cell sorter drop frequency was 37 kHz, stream focus was 10%, maximum drop charge was 110 V. A six-way sorting, 1.0 drop pure sort mode was used to sort the cell subpopulations. Stream deflections were −84, −65, −32, 32, 65, and 84 for six-way sort from left to right. For each participant, 1 million cells (Cocktail 1 panel) were run first to calculate the potential yield of each subpopulation, including neutrophils, B lymphocytes, monocytes, NK cells, naïve CD4⁺ and CD8⁺, CM CD4⁺ and CD8⁺, EM CD4⁺ and CD8⁺, and TEMRA CD4⁺ and CD8⁺ lymphocytes (total cell number × percentage of each subpopulation). The six most abundant subpopulations were sorted. Purity checks were performed on all sorted subpopulations to ensure the instrument performance was good enough to reach the sorted population purity >95%. Raw data (.fcs file) was exported for further analysis on FCS Express 7 research version.

## Processing and storage of sorted cells

Following flow cytometry, sorted cell subtypes were transferred and pooled by pipetting about half of each collection tube (2.5 ml) into larger falcon tubes, gently vortexing to liberate cells that may have adhered to the tube wall, and the remaining volume pipetted into the transfer tube. HBSS/ 2% BSA was used as necessary to equilibrate and cells were centrifuged at 1000×$g$ for 5 min. Following centrifugation, each cell pellet was isolated by gently decanting the supernatant and re-suspended into 1 ml of HBSS/ % BSA. The resulting purified cell suspensions were transferred to a 1.5-ml Eppendorf tube for each cell type, centrifuged at 2000×$g$ for 2 min at 4°C, and the supernatant carefully removed to leave a dry cell pellet. Samples were stored in liquid nitrogen for 4–12 months (– 170°C) until mitochondrial biochemistry and mtDNAcn analyses were performed as a single batch.

## Mitochondrial enzymatic activities

Samples were thawed and homogenized in preparation for enzymatic activity measurements with one tungsten bead and 500 µl of homogenization buffer (1 mM EDTA, 50 mM Triethanolamine). Tubes were transferred to a pre-chilled rack and homogenized using a Tissue Lyser (Qiagen, Cat# 85300) at 30 cycles/s for 1 min. Samples were then incubated for 5 min at 4°C, and homogenization was repeated for 1 min and the samples were returned to ice ready for enzymatic assays.

Mitochondrial enzyme activities were quantified spectrophotometrically for CS, COX (Complex IV), SDH (Complex II), and NADH dehydrogenase (Complex I) as described previously (*Picard et al., 2018*) with minor modifications. Each sample was measured in triplicates for each enzymatic assay (three wells for total activity and three wells for non-specific activity, except for the COX assay where a single non-specific activity value is determined across 30 wells). Homogenate volumes used for each reaction were: CS: 10 µl, COX and SDH: 20 µl, and Complex I: 15 µl.

CS activity was measured by detecting the increase in absorbance at 412 nm, in a reaction buffer (200 mM Tris, pH 7.4) containing acetyl-CoA 0.2 mM, 0.2 mM 5,5'- dithiobis-(2-nitrobenzoic acid) (DTNB), 0.55 mM oxaloacetic acid, and 0.1% Triton X-100. Final CS activity was obtained by integrating OD412 change from 150 to 480 s, and by subtracting the non-specific activity measured in the absence of oxaloacetate. COX activity was measured by detecting the decrease in absorbance at 550 nm, in a 100 mM potassium phosphate reaction buffer (pH 7.5) containing 0.1% n-dodecylmaltoside and 100 µM of purified reduced cytochrome c. Final COX activity was obtained by integrating OD550 change over 200–600 s and by subtracting spontaneous cyt c oxidation without cell lysate. SDH activity was measured by detecting the decrease in absorbance

at 600 nm, in potassium phosphate 100 mM reaction buffer (pH 7.5) containing 2 mM EDTA, 1 mg/ml BSA, 4 µM rotenone, 10 mM succinate, 0.25 mM potassium cyanide, 100 µM decylubuquinone, 100 µM DCIP, 200 µM ATP, and 0.4 µM antimycin A. Final SDH activity was obtained by integrating $OD^{600}$ change over 200–900 s and by subtracting activity detected in the presence of malonate (5 mM), a specific inhibitor of SDH. Complex I activity was measured by detecting the decrease in absorbance at 600 nm, in potassium phosphate 100 mM reaction buffer (pH 7.5) containing 2 mM EDTA, 3.5 mg/ml BSA, 0.25 mM potassium cyanide, 100 µM decylubuquinone, 100 µM DCIP, 200 µM NADH, and 0.4 µM antimycin A. Final Complex I activity was obtained by integrating $OD^{600}$ change over 120–600 s and by subtracting activity detected in the presence of rotenone (500 µM) and piericidin A (200 µM), specific inhibitors of Complex I. All assays were performed at 30°C. The molar extinction coefficients used were 13.6 L mol$^{-1}$ cm$^{-1}$ for DTNB, 29.5 L mol$^{-1}$ cm$^{-1}$ for reduced cytochrome c, and 16.3 L mol$^{-1}$ cm$^{-1}$ for DCIP to transform change in OD into enzyme activity.

Mitochondrial enzymatic activities were measured on a total of 340 samples, including 136 replicates of the same cell type for the same person. This provided more stable estimates of enzymatic activities than single measures would for a total of 204 individual person-cell combinations. The technical variation for each enzyme varied according to cell type, with cell types with lower enzymatic activities generally showing the highest CV. CV averaged across all cell types were: CS=6.3%, Complex I=16.6%, SDH=9.3%, and COX=23.4 % (*Supplementary file 3*).

mtDNAcn was determined as described previously (*Picard et al., 2018*) with minor modifications. The same homogenate used for enzymatic measurements (20 µl) was lysed in lysis buffer (100 mM Tris HCl pH 8.5, 0.5 % Tween 20, and 200 µg/ml proteinase K) for 10 hr at 55°C followed by inactivation at 95°C for 10 mi. About 5 µl of the lysate was directly used as template DNA for measurements of mtDNA copy number. qPCR reactions were set up in triplicates using a liquid handling station (ep-Motion5073, Eppendorf) in 384-well qPCR plates. Duplex qPCR reactions with Taqman chemistry were used to simultaneously quantify mitochondrial and nuclear amplicons in the same reactions. *Master Mix*1 for ND1 (mtDNA) and B2M (nDNA) included: TaqMan Universal Master mix fast (Life Technologies #4444964), 300 nM of primers and 100 nM probe (ND1-Fwd: GAGCGATGGTGAGAGCTAAGGT, ND1-Rev: CCCTAAAACCCGCCACATCT, Probe:HEX-CCATCACCCTCTACATCACCGCCC-3IABkFQ.B2M-Fwd:CCAGCAGAGAATGGAAAGTCAA,B2M-Rev: TCTCTCTCCATTCTTCAGTAAGTCAACT, Probe:FAM-ATGTGTCTGGGTTTCATCCATCCGACA-3IABkFQ). *Master Mix*2 for COX1 (mtDNA) and RnaseP (nDNA) included: TaqMan Universal Master Mix fast, 300 nM of primers and 100 nM probe (COX1-Fwd: CTAGCAGGTGTCTCCTCTATCT, COX1-Rev: GAGAAGTAGGACTGCTGTGATTAG, Probe: HEX-TGCCATAACCCAATACCAAACGCC-3IABkFQ. RnaseP-Fwd: AGATTTGGACCTGCGAGCG, RnaseP-Rev: GAGCGGCTGTCTCCACAAGT, Probe: FAM-TTCTGACCTGAAGGCTCTGCGCG-3IABkFQ). The samples were then cycled in a QuantStudio seven flex qPCR instrument (Applied Biosystems, Cat# 4485701) at 50°C for 2 min, 95°C for 20 s, 95°C for 1 min, 60°C for 20 s for 40× cycles. Reaction volumes were 20 µl. To ensure comparable Ct values across plates and assays, thresholds for fluorescence detection for both mitochondrial and nuclear amplicons were set to 0.08.

mtDNAcn was calculated using the ΔCt method. The ΔCt was obtained by subtracting the average mtDNA Ct values from the average nDNA Ct values for each pair ND1/B2M and COX1/RNaseP. Relative mitochondrial DNA copies are calculated by raising 2 to the power of the ΔCt and then multiplying by two to account for the diploid nature of the nuclear genome (mtDNAcn=2ΔCt×2). Both ND1 and COX1 yielded highly correlated mtDNAcn and the average of both amplicon pairs was used as mtDNAcn value for each sample. The overall CV across all cell subtypes was 5.1% for mtDNAcn.

## Platelet depletion in PBMCs

For this experiment, participants were nine community-dwelling older adults (mean age=79, range: 64–89, 4 women and 5 men). The sample included 7 White and 2 African American participants. Exclusion criteria included diseases or disorders affecting the immune system including autoimmune diseases, cancers, immunosuppressive disorders, or chronic, severe infections; chemotherapy or radiation treatment in the 5 years prior to enrollment; unwillingness to undergo venipuncture; immunomodulatory medications including opioids and steroids; or more than two of the following classes of medications: psychotropics, anti-hypertensives, hormones replacement, or thyroid supplements. Participants were recruited from a volunteer subject pool maintained by

the University of Kentucky Sanders-Brown Center on Aging. The study was conducted with the approval of the University of Kentucky Institutional Review Board. Blood (20 ml) was collected by venipuncture into heparinized tubes in the morning hours to control for potential circadian variation. PBMCs were isolated from diluted blood by density gradient centrifugation (20 min at 800×$g$, brake off) using Histopaque (Sigma-Aldrich, St. Louis, MO). Buffy coats were washed once, and cells were counted using a hemocytometer. PBMCs (20–30 M) were cryopreserved in liquid nitrogen in RPMI-1640 (Lonza) + 10% FBS (Hyclone) + 10% DMSO (Thermo Fisher Scientific), until further processing.

For platelet depletion, PBMCs were first thawed at room temperature and centrifuged at 500×$g$ for 10 min. The supernatant was discarded, and the cells were resuspended in 2 ml of HBSS without phenol red, calcium and magnesium (Life Technologies, Cat#14175103). Cells were then counted on the Countess II FL Automated Cell Counter (Thermo Fisher Scientific, Cat# AMQAF1000) in a 1:1 dilution with trypan blue. Cells were then divided into two aliquots: (1) 10 million cells for total PBMCs; (2) 11 million cells for platelet depletion. The total PBMCs were centrifuged at 2000×$g$ for 2 min at 4°C and subsequently frozen as a dry pellet at – 80°C until processing for enzymatic assays and qPCR. The PBMCs destined for platelet deletion cells were processed immediately.

The total PBMCs cell preparation was first immunolabeled with magnetically coupled antibodies against the platelet marker CD61. The 11 million platelet-depleted PBMCs were then centrifuged at 300×$g$ for 10 minutes. After spin, the supernatant was aspirated and cells were resuspended in 80 μl of HBSS. Then 20 μl of the CD61 MicroBeads (Miltenyi Biotec, Cat# 130-051-101) were added to the cells and incubated for 15 min at 4°C to magnetically label platelets. Cells were washed with 2 ml of HBSS and centrifuged at 300×$g$ for 10 min. The LS column (Miltenyi Biotec, Cat# 130-042-401) was placed in the magnetic field of the MACS Separator (Miltenyi Biotec, Cat# 130-091-051). The LS column was equilibrated with HBSS, cells resuspended in 500 μl of HBSS were applied to the LS column, and the CD61- cells were flown through the column and collected in a 15-ml collection tube. The LS column was then washed 3× with 500 μl of HBSS. The flow through was then spun at 500×$g$ for 10 min, the cell pellet was resuspended in 2 ml of HBSS and re-counted to isolate 10 M platelet-depleted cells. These cells were pelleted at 2000×$g$ for 10 min at 4°C, the supernatant removed, and cell pellet stored at – 80°C. The platelets (CD61$^+$) were recovered by flushing 1 ml of HBSS trhough the LS column with the plunger in a new tube, centrifuged at 3000×$g$ for 10 min, the supernatant removed, and the cell pellet stored at – 80°C until all samples could be processed for enzymatic activity assays as a single batch. For each participant, this experiment yielded three samples: (1) total PBMCs, (2) platelet depleted PBMCs, and (3) enriched platelets. Each sample was processed in parallel for RC enzymatic activity assays and mtDNAcn as described above.

## Mitotypes

As in other complex biological systems, the function and behavior of mitochondria require synergy among multiple features. Therefore, the functional characteristics of mitochondria can be expressed by multivariate indices reflecting the inter-relations of multiple mitochondrial features. This logic is similar to body mass index (BMI), which integrates height and weight into a BMI ratio that is more easily interpretable and meaningful than either height or weight features alone. In relation to mitochondria, in human tissues, the ratio of CIV and CII activities (also known as COX/SDH ratio) is used in diagnoses of mitochondrial dysfunction, whereas alone either CIV or CII activities are less easily interpretable (*Picard et al., 2012*). Thus, integrating primary features of mitochondrial content, genome abundance, and RC function produces integrated *mitotypes*, which reflect the functional specialization of mitochondria among different cell subtypes, or among dynamic cellular states.

Similar to BMI, the mitotypes analyzed (listed and defined in *Figure 7—figure supplement 1*) are computed from the ratio of two or more mitochondrial features. Each mitotype can be visualized as a scatterplot with two variables of interest as the x and y axes (*Figure 7a*). In the example in *Figure 7a*, the hypothetical cell type A has a higher value for feature two relative to feature 1 (Mitotype A), while cell type B has a higher value for feature one relative to feature 2 (Mitotype B). Note that if both features increase or decrease following the same proportions, the ratio is the same and follows a diagonal on the mitotype graph, reflecting an invariant mitotype. Thus, similar to BMI, changes in mitotypes are reflected by perpendicular movement relative to the diagonal in the mitotype space.

## Statistical analyses

To adjust for potential order effects across the 340 samples (31 samples per 96-well plate, 17 plates total) a linear adjustment was applied to raw values enzymatic activity measures, which adjusts for potential storage and batch effects, ensuring consistency between the first and last samples assayed. Samples from both the cohort and the repeat participant were processed and analyzed as a single batch.

Mann-Whitney t-tests were used to compare sex differences in cell type proportions and mitochondrial measures. Throughout, effect sizes between groups were computed as Hedges' g (g) to quantify the magnitude of group differences in cell type proportions and mitochondrial measures (by sex, mitotype cell subtype, and inter-individual differences). Spearman's r (r) was used to assess the strength of associations between continuous variables such as age and cell proportion or age and mitochondrial measures. To assess to what extent mitochondrial features are correlated across cell subtypes (co-regulation) and to calculate the average correlation across mitotypes, Spearman's r matrixes were first computed and transformed to Fisher's Z', and then averaged before transforming back to Spearman's r ($r_{z'}$). One-way nonparametric ANOVA Kruskal-Wallis with post hoc Dunn's multiple comparison tests were used to compare cell type mitochondrial measures in different cell subtypes and PBMCs. Between- and within-person variation were characterized using CVs. The root mean square of successive differences (rMSSD) was computed to quantify the magnitude of variability between successive weeks for repeated measures. Chi-square tests were computed to compare the proportion of mitotype indices categories (enzyme activity per CS, enzyme ratios, enzyme per mtDNA, enzyme per mtDNA density, and enzyme per mtDNA relative to mtDNA density) by age (lower vs higher with increased age) and sex (lower vs higher in men). Finally, one-way nonparametric Friedman tests with post hoc Dunn's multiple comparisons were used to compare mitochondrial measures in platelet-depleted PBMCs, enriched platelets PBMCs, and total PBMCs. Statistical analyses were performed with Prism 8 (GraphPad, CA), R version 4.0.2, and RStudio version 1.3.1056. Statistical significance was set at $p < 0.05$.

## Appendix 2

### Mitochondrial health index among cell subtypes

To define how RC function relates to mitochondrial content, we integrated content and function features into a single metric known as the MHI. The MHI reflects *RC capacity on a per-mitochondrion basis* (*Appendix 2—figure 1*). In this study, the MHI was adapted from *Picard et al., 2018* with the addition of RC complex I (CI) activity.

Normalizing RC activities to mitochondrial content (i.e., CS) is considered the gold standard to evaluate RC dysfunction (*Frazier et al., 2020*). But given the limitations of individual mitochondrial content markers for specific cell types (*Larsen et al., 2012*; *McLaughlin et al., 2020*; *Picard, 2021*), integrating CS and mtDNAcn into a single factor (denominator: CS+mtDNAcn) must yield a more generalizable estimate of mitochondrial content. Similarly, there may not be a single best marker of RC function across all cell types, and the RC chain operates as a multi-enzyme system that depends on all complexes. This provides the same rationale to integrate CI, CII, and CIV into a single factor (numerator: CI+CII + CIV). Integrating individually measured metrics into latent constructs also has the added advantage of triangulating measurement error, thus yielding more stable quantitative estimates of mitochondrial content and RC function. In three recent studies of chronic life stress in human PBMCs (*Picard et al., 2018*), ovarian tumor inflammation (*Bindra et al., 2021*), and mouse behavior (*Rosenberg et al., 2021*), we found more robust associations using MHI than individual metrics of content and RC function alone. This suggests that MHI is a functionally relevant metric of mitochondrial biology.

The resulting MHI metric is directly interpretable. An MHI value of 100 represents the average RC activity per mitochondrion in the whole data set. MHI values<100 reflect lower energy production capacity, whereas >100 reflect higher energy production capacity per unit of mitochondrion. For example, an individual with monocytes MHI of 80 has 20% lower RC function per unit of mitochondria than the dataset average; whereas someone with an MHI of 150 is 50% above the mean.

In our purified leukocyte data set, MHI significantly differed between cell subtypes (ANOVA, p<0.0001). Monocytes had the highest MHI, which was 72% higher than B cells (g=3.78, p<0.0001), which in turn had the lowest MHI (*Appendix 2—figure 1*). On average, CD8[+] memory T cells exhibited a 6–18% higher MHI relative to their naïve precursor (g=1.02, p<0.05), consistent with the notion that naïve and activated immune cells have different bioenergetic requirements (*Nicoli et al., 2018*; *van der Windt et al., 2012*). The MHI exhibited moderately consistent coherence between cell subtypes, falling in between the more coherent mitochondrial content markers (CS and mtDNAcn) and the less coherent RC markers (CI, CII, CIV) (*Appendix 2—figure 1*).

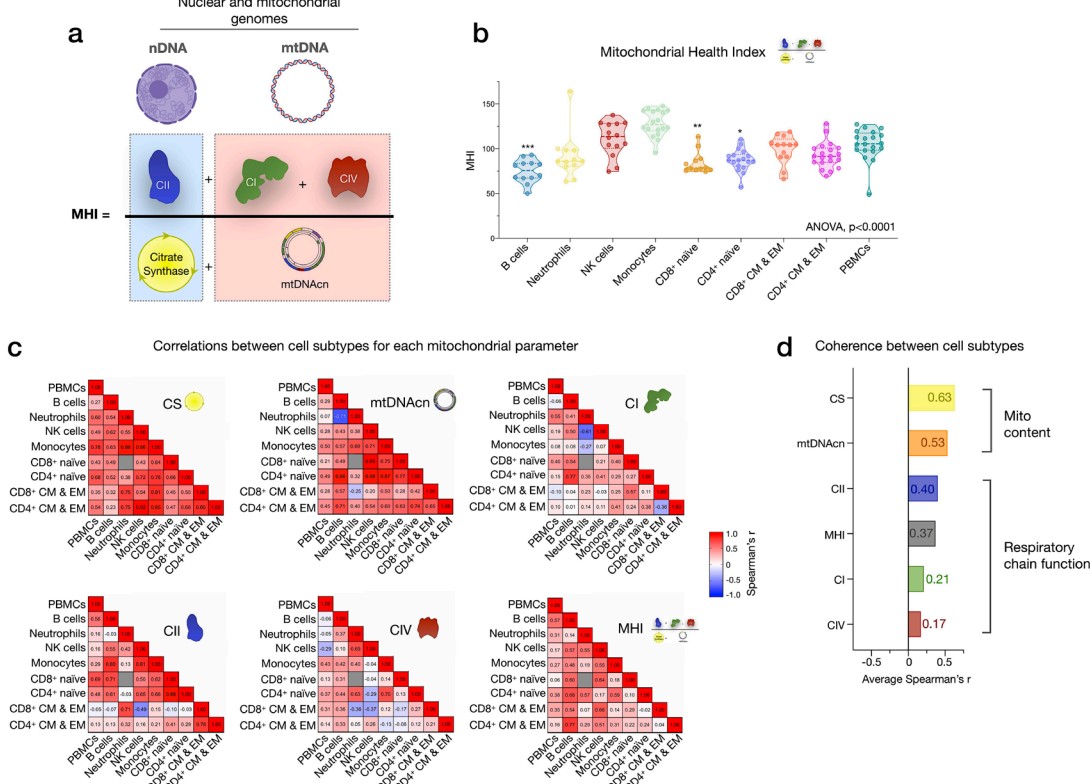

**Appendix 2—figure 1.** Mitochondrial health index (MHI) and coherence of mitochondrial features across cell subtypes. (**a**) Schematic of the MHI equation reflecting respiratory chain function as the numerator, and markers of mitochondrial content as the denominator, yielding a metric of energy production capacity on a per-mitochondrion basis. (**b**) MHI across immune cell subtypes. Dashed lines are median (thick), with 25th and 75th quartiles (thin). p-values from one-way nonparametric ANOVA Kruskal-Wallis test with Dunn's multiple comparison test of subtypes relative to PBMCs, n=12–18 per cell subtype. (**c**) Correlation matrices (Spearman's r) showing the association between cell subtypes in mitochondrial features. Correlations were not computed for cell subtype pairs with fewer than n=6 observations (gray cell). (**d**) Average effect sizes reflecting the within-person coherence of mitochondrial features across cell types (calculated using Fisher's z-transformation). p<0.05*, p<0.001***.

The online version of this article includes the following source data for appendix 2—figure 1:

- **Appendix 2—figure 1—source data 1.** Mitochondrial health index and coherence of mitochondrial features across cell subtypes.

