## [Editor Report]

In this manuscript the authors have assessed age-, sex- and time-driven differences in mitochondrial phenotypes with human peripheral blood mononuclear cells and their subsets. They find that differences in metabolic profile can be driven by differences in leukocyte composition (driven by age, sex, other stimuli) and platelet contamination. The leads obtained from this study would be useful for further research.

---

## [Decision Letter]

**Decision letter after peer review:**

Thank you for submitting your article "Mitochondrial phenotypes in purified human immune cell subtypes and cell mixtures" for consideration by *eLife*. Your article has been reviewed by 3 peer reviewers, and the evaluation has been overseen by a Reviewing Editor and Matt Kaeberlein as the Senior Editor. The reviewers have opted to remain anonymous.

This paper is of broad interest to those in the field of immunometabolism, a field which has largely used the mouse as an experimental system. The descriptive work is the first of its kind to demonstrate important aspects of biologic variability in and hidden aspects of mitochondrial function in human immune cells, and the data provided answers some important questions, while also generating new intriguing hypotheses. Overall, there is high enthusiasm for this study. The reviewers noted some issues which need to be addressed prior to publication. These are detailed below. In particular, limitations of this study that should be discussed at length in a revised manuscript. For example, the consensus in review was that the sample size is sufficient for a novel report with important, mainly descriptive, findings, but the manuscript must discuss in detail the need for much larger cohorts in future studies and the importance of future efforts to appropriately stratify human data by common variables when exploring these findings in greater detail. In general, however, the interest in this study is high, and the scope and magnitude of work associated with it appreciated, and we hope to you will be submitting a revised version addressing the issues noted. Finally, two of the reviewers note that the importance of leukocyte composition was already clear, so any claims of novelty in demonstrating that bulk PBMC measures are of limited value should be tempered (though there is value in highlighting this fact).

Essential revisions:

1) A major finding of this manuscript was that metabolic analysis of bulk PBMC masks a lot of differences that occur due to other factors, especially changes in leukocyte composition. From the perspective of an expert in immuno-metabolism, it is already very clear that i) different leukocytes have very different metabolic profiles and ii) the metabolic profile of a defined leukocyte subset can also change due to age-, infection-related changes in both composition and cell activation. Accordingly, analysis of bulk PBMCs will have limited value for defining biological mechanisms due to the complexity of cell subsets within that tissue sample. Perhaps researchers outside immunology are not aware of the complexity inherent in analysis of PBMCs, so there may be value in highlighting these differences, but that finding was of limited novelty to experts in the field.

2) Figure 2: there seems to be a disconnect between the text and the figures in terms of how the stats are talked about. Also, there is some data interpretation that is left to the reader, rather than spelling it out. Please revise to clarify.

3) Figure 4: the discrepancy between CD and mtDNAcn is concerning when considered against the literature – for many years people used one or the other or both together to refer to mito copy number (mito content). It is clear from your data that some cells show a disconnect, an important observation. Please provide a brief discussion of this finding.

4) Reviewers, and the reviewing editor, agree that while a 'per-mitochondria' normalized value may provide some insights, the 'mitochondrial health index,' or 'MHI', lacks biological meaning or interpretability. Summing individual ETC rates and dividing the sum by the sum of citrate synthase and mtDNA provides a number, but how does it relate to any discrete biologic property? It is easy to imagine interesting changes being masked (perhaps CI and II inversely change in a sample, nullifying either effect), or disproportionately weighed (CS and mtDNA, while here shown to not be equivalent, should still generally move together). Please either replace these data with an interpretable value or values (perhaps multiple individual values comparing single ETC rates to mtDNA) or move them to figure supplement.

5) Figure 6: In here and other places, it is difficult to parse apart what is statistically significant and what are trends. An asterix (significance) appears in some places, but not in others where differences are being highlighted. Is figure 6g, mtDNAcn in CD8^+^ T-cells EM and CM really different for CS? Please annotate figures with statistical values accordingly.

6) The value of figure 6 is a bit murky, as the findings are not sufficiently expanded upon and proper statistical methodology not indicated (such as multiple testing correction). It is not critical to the story. If included, it should be modified accordingly. Retaining this data for a future study may be appropriate.

7) One major variable that has not been assessed is variability in processing of bloods. The authors highlight that there is marked variation in samples collected from the same individual week to week but also highlight that platelet contamination can have a major impact on the readouts and that the range of variation in the age/sex cohort is similar to an individual's variation. So does the variability in the mitochondrial assays reflect variation in processing? It would be instructive to sample from the same individual sequentially over 5 days or even the same individual on the same day, process separately, and then perform these assays.

8) Figure 1: It does not appear that CM and EM are defined in the text, please address.

9) Figure 3 b-i: it is unclear why CD8 memory phenotype cells are missing from g-i, although they are present in the graphs b-e?

10) Supp Figure 2: On the CD4 vs CD8 gate, the gates are mislabelled as CCD4 and CCD8

11) It is unclear on how PBMCs were processed for all of the workflows. In particular, were the cells frozen and then defrosted at any point? Some PBMC subsets will withstand this process better and PBMCs may require resting time after thawing to return to normal metabolic activity.

*Reviewer #1 (Recommendations for the authors):*

This is a nice contribution to the field of immunometabolism, the first of its kind for human immune cell populations. A few suggestions for improvement.

Figure 1: It does not appear that CM and EM are defined in the text.

Figure 2: there seems to be a disconnect between the text and the figures in terms of how the stats are talked about. Also, there is some data interpretation that is left to the reader, rather than spelling it out.

Figure 3: the contribution of platelets to the observed phenotypes is very useful and serves as a cautionary tale.

Figure 4: the discrepancy between CD and mtDNAcn is concerning. For many years people used one or the other or both together to refer to mito copy number (mito content). It is clear from your data that some cells show a disconnect. How does that factor into further calculations such as the MHI? Is CS + mtDNAcn really mito content? What then is the MHI really telling us? Especially since the original description of the MHI was for leukocytes, clearly a heterogeneous population according to your data. Is there a gold standard to compare the MHI to?

Figure 5: How do you define mitochondrial content when you have discrepant CS and mtDNAcn?

Figure 6: In here and other places, it is difficult to parse apart what is statistically significant and what are trends. An asterix (significance) appears in some places, but not in others where differences are being highlighted. Is figure 6g, mtDNAcn in CD8^+^ T-cells EM and CM really different for CS?

Figure 10: Is this figure necessary? I am not sure it adds to your story. Are you trying to tell to many stories?

Just overall, since you did do flow, I was surprised not see some flow measurements of mito parameters and correlation with other outcomes. However, I do appreciate the scale of this study and the work involved.

*Reviewer #2 (Recommendations for the authors):*

The methods identify the last author as the donor for the repeated measures analysis. I understand that the author consents but I'm not sure if this is consistent with *eLife*'s policy with regard to human ethics, so I defer to that policy/practice but wanted to flag this. My sense is that it could be an undesirable precedent/practice to identify the biological/health data of authors, even with their consent, and it is unnecessary information for interpretation of the work.

*Reviewer #3 (Recommendations for the authors):*

While the question is worthwhile asking, the sample size is too small to reach any conclusions due to the inherent heterogeneity of the group. The major observation is the discrepancy observed between the mitochondria phenotype of bulk PBMCs and isolated subsets and it is somewhat expected. Doing this in human is complicated due to the interindividual variation and sample size has to be much bigger to properly stratify the data among age groups, sexes, psychiatric disorders that the subjects may have, medications, lifestyle factors such as BMI, exercise, stress levels etc. In general, bulk PBMCs can never be accepted as a representative for the immune subsets even for mouse studies. In its original form, the data can serve as a resource for conducting assays related to mitochondrial profiling. But the hypothesis is not original and there are many confounding factors such as the heterogeneity of the cohort and insufficient sample size for the time-course study. While the findings may provide leads for future studies in human immune subsets, this reviewer doesn't recommend its publication in *eLife*.

[Editors' note: further revisions were suggested prior to acceptance, as described below.]

Thank you for resubmitting your work entitled "Mitochondrial phenotypes in purified human immune cell subtypes and cell mixtures" for further consideration by *eLife*. Your revised article has been reviewed by 3 peer reviewers and the evaluation has been overseen by Anna Akhmanova as the Senior Editor, and a Reviewing Editor.

The manuscript has been improved but there are some remaining issues that need to be addressed, as outlined below:

While the reviewers were generally satisfied with the revisions, reviewers 2 and 3 and the reviewing editor feel there are issues that need to be addressed prior to acceptance.

1. MHI. Reviewers 2 and 3 and the reviewing editor all find the added description and citations supporting the MHI to be insufficient to allay concerns over the lack of biological meaning, functional relevance, or interpretability of this measure. This analysis is not important to any of the key findings of this study, so we ask that Figure 5 and related text be moved to supplemental data.

2. Reviewers request further explicit acknowledgement that the distinct metabolic phenotypes for distinct cell populations is already well known in immunology. In your response, please clearly highlight text addressing this concern.

*Reviewer #1 (Recommendations for the authors):*

The authors have adequately addressed my concerns and suggestions.

*Reviewer #2 (Recommendations for the authors):*

The authors have responded to comments from the reviewers but I would have liked to have seen more engagement on some items.

Specifically, two reviewers noted that they would have expected analysis of another more commonplace measure of mitochondrial metabolism alongside MHI. While I am aware of the short-comings of other methods, and cogniscient that it would be impractical to do at the same time as the sorting experiments that have been performed, a parallel analysis with a small subset of unsorted cells would be feasible. I view this kind of analysis as important to benchmark this MHI metric relative to other work in the field.

All of the reviewers wanted to see more acknowledgement that the distinct metabolic phenotypes for distinct cells is well known in immunology and I see some changes but it is hard to gauge the extent of the changes. Also all of the reviewers wanted to see more tempered language for the statistical analyses and again I see some changes but hard to gauge. I can't see a track-changed copy in the documents but it would have been helpful to better assess how the language has changed.

The framing especially in the first couple of paragraphs of the introduction is still a bit unfocused and the challenges that the authors are aiming to address (validating an integrated measure of mitochondrial health for large translational studies, exploring the application of that measure with PBMCs that are a complex mix of cells) are not well described at the beginning of the abstract.

*Reviewer #3 (Recommendations for the authors):*

The authors have successfully addressed most of the issues raised during initial review. Although reservations about the importance of the main hypothesis and sample size remain, this reviewer thinks that the methodology and results may provide valuable information and a reference point for an audience who wishes to measure mitochondrial parameters in human PBMC subsets. Therefore the final decision is at the editor's discretion.

---

## [Author Response]

Essential revisions:1) A major finding of this manuscript was that metabolic analysis of bulk PBMC masks a lot of differences that occur due to other factors, especially changes in leukocyte composition. From the perspective of an expert in immuno-metabolism, it is already very clear that i) different leukocytes have very different metabolic profiles and ii) the metabolic profile of a defined leukocyte subset can also change due to age-, infection-related changes in both composition and cell activation. Accordingly, analysis of bulk PBMCs will have limited value for defining biological mechanisms due to the complexity of cell subsets within that tissue sample. Perhaps researchers outside immunology are not aware of the complexity inherent in analysis of PBMCs, so there may be value in highlighting these differences, but that finding was of limited novelty to experts in the field.

Indeed, some researchers outside of immunology are generally either not aware of the complexity inherent in analysis of PBMCs, or they do not think those differences are substantial enough to influence mitochondrial measurements. We have revised the manuscript to remove any claim of novelty and to better contextualize our findings related to PBMCs. The contribution of this manuscript is to quantify the magnitude of these mitochondrial differences between circulating human immune cell subtypes, and to quantitatively demonstrate their influence on PBMC-based mitochondrial measurements. We agree that there is value in highlighting these differences given that the target audience is not only immunologists but also translational scientists who frequently use PBMCs out of convenience (because they are easier to obtain in high numbers than specific cell subtypes). We hope that the data in this manuscript will provide the needed empirical basis to design future human/clinical studies with adequate cell type specificity.

2) Figure 2: there seems to be a disconnect between the text and the figures in terms of how the stats are talked about. Also, there is some data interpretation that is left to the reader, rather than spelling it out. Please revise to clarify.

We apologize for the confusion. Figure 2a shows the Spearman (r) correlation coefficients, whereas the text discusses the squared correlation coefficient (r^2^). We previously chose to discuss r^2^ in the manuscript because it reflects the proportion of shared variance between two variables and is therefore more directly interpretable. We have revised this section of the results to clarify, which now reads as follows:

“Notably, the correlation between circulating B cell abundance and PBMCs CS activity was r=0.78 (p<0.0001), meaning that the proportion of shared variance (r^2^) between both variables is 61% (i.e., B cell abundance explains 61% of the variance in PBMCs CS). Similarly, the correlations between B cell abundance and PBMCs mtDNAcn, CI, CII activities were 0.52-0.67 (27-45% of shared variance, ps<0.05-01).”

The other correlations in the table did not reach statistical significance so are not discussed further, except for the particularly consistent pattern of correlations with memory CD4^+^ and CD8^+^ subtypes. The individual scatterplots for each correlation are also provided in Figure 2b for readers who wish to examine these correlations in more details.

3) Figure 4: the discrepancy between CD and mtDNAcn is concerning when considered against the literature – for many years people used one or the other or both together to refer to mito copy number (mito content). It is clear from your data that some cells show a disconnect, an important observation. Please provide a brief discussion of this finding.

Thank you for this comment. We agree that CS and mtDNAcn are generally both considered valid markers of mitochondrial content across tissue types. However, we recently performed a careful review of the primary evidence for mtDNAcn as a marker of mitochondrial content and find that contrary to popular belief mtDNAcn can change independently of mitochondrial mass or content in different tissues (Picard, 2021). For example, one study in human skeletal muscle used electron microscopy to quantify mitochondrial content in parallel with CS activity and mtDNAcn, which showed that CS activity was the most highly correlated with mitochondrial content (r=0.84, p<0.001) whereas mtDNAcn showed a markedly lower and non-significant correlation (r=0.35, p=0.23) (Larsen et al., 2012). Nevertheless, the correlation of mtDNAcn and mitochondrial content was positive, and it may be more considerable in other tissues. Therefore, although the evidence for using mtDNAcn is mixed, we are not prepared to completely abandon mtDNAcn as a marker of mitochondrial content. Similarly, we cannot be certain that CS is a reliable marker of mitochondrial content across all cell and tissue types, and across different conditions.

For these reasons, our current view is that using two orthogonal markers (mtDNAcn, CS) for the same construct (i.e., mitochondrial content) is likely to yield a more accurate estimate than either alone. This logic also constitutes part of the rationale for integrating these independent measurements into a single index (such as the MHI). As requested, we now discuss this observation in the Results section:

“The correlations between different mitochondrial features indicate that CS and mtDNAcn were only weakly correlated with each other, and in some cases were negatively correlated (Figure 4f). This finding may be explained by the fact that although both CS and mtDNAcn are positively related to mitochondrial content, CS may be superior in some cases (Larsen et al., 2012), inadequate in specific tissues (McLaughlin et al., 2020), and that mtDNAcn can change independently of mitochondrial content and biogenesis (Picard, 2021).”

4) Reviewers, and the reviewing editor, agree that while a 'per-mitochondria' normalized value may provide some insights, the 'mitochondrial health index,' or 'MHI', lacks biological meaning or interpretability. Summing individual ETC rates and dividing the sum by the sum of citrate synthase and mtDNA provides a number, but how does it relate to any discrete biologic property? It is easy to imagine interesting changes being masked (perhaps CI and II inversely change in a sample, nullifying either effect), or disproportionately weighed (CS and mtDNA, while here shown to not be equivalent, should still generally move together). Please either replace these data with an interpretable value or values (perhaps multiple individual values comparing single ETC rates to mtDNA) or move them to figure supplement.

We agree that individual measures are valuable and should not be excluded from the interpretation. For this reason, we present each individual measure individually in Figure 4. We subsequently introduce the MHI as a tool to capture total RC activity on a per-mitochondria basis, which we also agree can provide insight, but was insufficiently described in the previous version of the manuscript.

Normalizing mitochondrial ETC activities to mitochondrial content is the gold standard to evaluate ETC dysfunction (Frazier, Vincent, Turnbull, Thorburn, and Taylor, 2020), and can be directly compared to our MHI approach. For example, in mitochondrial disease where mtDNA-encoded ETC complexes I and IV activities can be severely impaired, we and others have observed dramatic, compensatory elevations in mitochondrial content (i.e., in muscle, 2-3-fold elevation in CS). The result is that the total tissue ETC activities can appear normal, but when normalized to mitochondrial content with CS, the ETC activity/CS ratio can be substantially reduced.

But as noted above, we cannot be certain that CS is the absolute ideal marker of mitochondrial content across all cell types. Therefore, there is added value to mathematically combine two orthogonal measures of the same construct into a single factor (e.g., CS + mtDNAcn for mitochondrial content; ETC complexes for global ETC activity). This approach has two main advantages: (i) it makes each construct more robust in the event that one of the metrics is sub-optimal (in a given cell type, for example) or that the rate limiting factor in ETC function depends on the combined activity of multiple enzymes, and also (ii) triangulates technical variation that is inherent to any laboratory measurement, making the resulting index more statistically robust.

This point is supported by prior evidence showing that integrating ETC measures as numerator and mitochondrial content measures as denominator can generate an index with better discriminatory potential than individual measures alone. We initially tested this approach in a study of chronic caregiving stress, testing how well (i) individual measures, (ii) ratios of ETC enzymes to CS, and (iii) the MHI can discriminate between control vs caregiving groups (chronic stress model) (Picard et al., 2018). The result, illustrated in Figure 1E from Picard et al., 2018, shows as expected that ETC/CS ratios are superior than individual measures. The MHI has the highest discriminatory potential. We also recently found that compared to individual enzymes, MHI was most strongly correlated with interleukin 6 (IL-6) in ovarian carcinoma (Bindra et al., 2021). And in the mouse brain, MHI was most correlated with anxiety and social behaviors compared to individual ETC enzymatic activities and mitochondrial content measures CS and mtDNAcn (Rosenberg et al., 2021). In these three cases, we attribute the added value of the MHI to (i) a more representative biological assessment of the ETC function relative to mito content, and (ii) triangulation of measurement error that may increase the robustness of MHI.

In regards to the interpretability of the MHI, the index is designed such that the individual or cell type with the average ETC activity per unit of mitochondrion is 100 units. So, for example, a sample with an MHI value of 120 can be interpreted as having 20% higher energy production capacity relative to the average sample; a sample with an MHI of 50 (such as the lowest participant for B cells in Figure 5b) exhibits half of the average energy production capacity in the whole dataset.

If the reviewers and editor still believe that Figure 5 on MHI should be moved to the supplement, we are happy to do so. In case they will consider a revised description of the MHI and of these results, we have revised the corresponding section of the manuscript as follows:

“To further define how RC function relates to mitochondrial content, we integrated content and function features into a single metric known as the MHI, reflecting RC capacity on a per-mitochondrion basis (Figure 5a). […] Furthermore, the resulting index is also directly interpretable since an MHI value of 100 represents the average RC activity per mitochondrion in the dataset. MHI values <100 reflect lower energy production capacity, whereas >100 reflect higher energy production capacity per unit of mitochondrion.”

5) Figure 6: In here and other places, it is difficult to parse apart what is statistically significant and what are trends. An asterix (significance) appears in some places, but not in others where differences are being highlighted. Is figure 6g, mtDNAcn in CD8^+^ T-cells EM and CM really different for CS? Please annotate figures with statistical values accordingly.

Thank you for this comment, we have revised the Results section to clearly distinguish what is statistically significant and what is not. The majority of comparisons in Figure 6 are not significant because of the limited sample size. For this reason, we have focused our analysis and interpretation based on the standardized effect sizes (e.g., Hedge’s g), which is not systematically influenced by sample size. As requested, we have ensured that all statistical details are noted in the figure legend and that all statistical values, denoted with asterisks, are included.

6) The value of figure 6 is a bit murky, as the findings are not sufficiently expanded upon and proper statistical methodology not indicated (such as multiple testing correction). It is not critical to the story. If included, it should be modified accordingly. Retaining this data for a future study may be appropriate.

The analysis in Figure 6 is admittedly underpowered and should be considered exploratory, as the study was designed to primarily compare cell subtypes, and not initially designed to address age and sex differences. In writing this manuscript, we considered retaining this data for a future study, but given the objective of this manuscript to provide a broad foundation for human leukocyte studies, these results were too striking and insightful to omit from this manuscript. For example, CS activity was higher in women than in men, consistently across 8 out of 8 cell subtypes. Individual statistical tests for each cell subtype do not capture the biological significance of this result (effectively replicated independently across all cell types examined). The expected null distribution if there was no sex difference would be 4 cell types with higher CS in women, 4 higher in men (4:4); based on a Chi-square test against the null hypothesis (4:4), the observed proportion (8:0) is significantly different from chance (p=0.0047). Similarly, contrary to what has been observed in cell mixtures and whole blood for about two decades, mtDNAcn in all cell types show *positive* correlations with age, except in the only cell type whose abundance declines dramatically with age, CD8^+^ naïve T cells. These cell type-specific results have changed how we think about designing future studies, and how we evaluate prior literature on age-related changes in mtDNAcn, for example.

Statistically, because of the exploratory basis of these analyses and small sample size, we used non-parametric tests (Mann-Whitney for women-men comparisons in Figure 6a-f, and Spearman r for age analysis in Figure 6g-I) and did not adjust for multiple comparisons. Although exploratory, these results are fairly novel and might represent important evidence to guide the rational design of larger hypothesis-driven mitochondrial human leukocyte studies.

In case the reviewing editor and reviewers agree, we have thoroughly revised this section of the manuscript to highlight the exploratory nature of these sex- and age-related analyses, and to better highlight their potential biological significance that deserve validation in larger studies. However, if the editors do not see added value in Figure 6, it can certainly be moved to the supplement, or removed entirely from the manuscript.

7) One major variable that has not been assessed is variability in processing of bloods. The authors highlight that there is marked variation in samples collected from the same individual week to week but also highlight that platelet contamination can have a major impact on the readouts and that the range of variation in the age/sex cohort is similar to an individual's variation. So does the variability in the mitochondrial assays reflect variation in processing? It would be instructive to sample from the same individual sequentially over 5 days or even the same individual on the same day, process separately, and then perform these assays.

This is an excellent question, and a challenging problem to address. We have carefully considered this problem. The procedure for collecting and processing blood plus the time required for FACS limits the number of samples that can be processed in a single day. From the time of collection to the freezing of cell pellets for later mitochondrial profiling, the total workflow requires approximately 10 hours. It would not be feasible to process and sort multiple blood samples in one day without requiring multiple cell sorting instruments (our core only has one 6-channel sorter). If we had more than one sorter, this would also introduce another parameter to consider.

In regards to daily repeated sampling, our previous study showed that mitochondrial enzymatic activities (and MHI) were maximally influenced by psychological states (i.e., mood) 12-24 hours prior to the blood draw, more so than compared to 48 and 72 hours prior (Picard et al., 2018). This data suggests that mitochondrial ETC function may vary, in a way influenced by mood and behavior, from day-to-day. Moreover, we have new unpublished evidence using soluble mitochondrial markers (i.e., mitokines) collected multiple times per day, over multiple weeks, which show that mitochondria-derived proteins and cell-free mtDNA exhibit substantial day-to-day variation. Therefore, attributing daily variation in the same individual over 5 sequential days to real biology or to technical variation in processing bloods would, at best, be imperfect.

To address this problem, we have (1) rigorously standardized all aspects of the protocol described in great details in the methods section, and (2) ensured that the time of blood processing (time from draw to isolation, from isolation to sorting, and from sorting to freezing) was the same for each participant. We have also carefully quantified the technical variation in other aspects of protocol, including mitochondrial phenotyping, reported in Supplementary file 3.

8) Figure 1: It does not appear that CM and EM are defined in the text, please address.

Thank you for bringing this to our attention. We have revised the manuscript to define central memory (CM) and effector memory (EM) cells.

9) Figure 3 b-i: it is unclear why CD8 memory phenotype cells are missing from g-i, although they are present in the graphs b-e?

We could only sort 6 cell types (the most abundant) for each participant. Because the 6 most abundant cell types were slightly different between individuals, this yielded 8 cell subtypes to be compared in the n=21 cohort. But when comparing the cohort with the repeat participant, comparisons can only be made with the 6 cell subtypes available in this individual (neutrophils, NK cells, monocytes, naïve and CM-EM CD4^+^ T cells, and naïve CD8^+^ T cells). Thus, the cohort-repeat participant comparison in g-i have neither B cells nor CD8^+^ memory T cells, since the only data available is from the cohort. We have revised the text and figure legend to better clarify the cell subtypes plotted in Figure 8g-i.

10) Supp Figure 2: On the CD4 vs CD8 gate, the gates are mislabelled as CCD4 and CCD8

Corrected.

11) It is unclear on how PBMCs were processed for all of the workflows. In particular, were the cells frozen and then defrosted at any point? Some PBMC subsets will withstand this process better and PBMCs may require resting time after thawing to return to normal metabolic activity.

Thank you for asking to clarify this important point. PBMCs and FACS sorted cells were frozen and subsequently processed as a single batch for mitochondrial phenotyping. All assays are performed on homogenized cell lysates so viability is not preserved. There is therefore no concern with freezing for these assays. We have revised the first paragraph of the Results section to clarify, and ensured that the Methods section clearly describes the PBMCs processing workflow.

Reviewer #1 (Recommendations for the authors):This is a nice contribution to the field of immunometabolism, the first of its kind for human immune cell populations. A few suggestions for improvement.Figure 1: It does not appear that CM and EM are defined in the text.

Corrected.

Figure 2: there seems to be a disconnect between the text and the figures in terms of how the stats are talked about. Also, there is some data interpretation that is left to the reader, rather than spelling it out.

We apologize for the confusion. Figure 2a shows the Spearman (r) correlation coefficients, whereas the text discusses the squared correlation coefficient (r^2^). We previously chose to discuss r^2^ in the manuscript because it reflects the proportion of shared variance between two variables and is therefore directly interpretable. We have revised this section of the results to clarify, which now reads as follows:

“Notably, the correlation between circulating B cell abundance and PBMCs CS activity was r=0.78 (p<0.0001), meaning that the proportion of shared variance (r^2^) between both variables is 61% (i.e., B cell abundance explains 61% of the variance in PBMCs CS). Similarly, the correlations between B cell abundance and PBMCs mtDNAcn, CI, CII activities were 0.52-0.67 (27-45% of shared variance, ps<0.05-01).”

The other correlations in the table did not reach statistical significance so are not discussed further, except for the particularly consistent pattern of correlations with memory CD4^+^ and CD8^+^ subtypes. The individual scatterplots for each correlation are also provided in Figure 2b for readers who wish to examine these correlations in more details.

Figure 3: the contribution of platelets to the observed phenotypes is very useful and serves as a cautionary tale.

Agreed.

Figure 4: the discrepancy between CD and mtDNAcn is concerning. For many years people used one or the other or both together to refer to mito copy number (mito content). It is clear from your data that some cells show a disconnect. How does that factor into further calculations such as the MHI? Is CS + mtDNAcn really mito content? What then is the MHI really telling us? Especially since the original description of the MHI was for leukocytes, clearly a heterogeneous population according to your data. Is there a gold standard to compare the MHI to?

Thank you for this comment. CS and mtDNAcn are generally both considered valid markers of mitochondrial content across tissue types. However, we recently performed a careful review of the primary evidence for mtDNAcn as a marker of mitochondrial content and find that contrary to popular belief mtDNAcn can change independently of mitochondrial mass or content in different tissues (Picard, 2021). For example, one study in human skeletal muscle used electron microscopy to quantify mitochondrial content in parallel with CS activity and mtDNAcn, which showed that CS activity was the most highly correlated with mitochondrial content (r=0.84, p<0.001) whereas mtDNAcn showed a markedly lower and non-significant correlation (r=0.35, p=0.23) (Larsen et al., 2012). Nevertheless, the correlation of mtDNAcn and mitochondrial content was positive, and it may be more considerable in other tissues. Therefore, although the evidence for using mtDNAcn is mixed, we are not prepared to completely abandon mtDNAcn as a marker of mitochondrial content. Similarly, we cannot be certain that CS is a reliable marker of mitochondrial content across all cell and tissue types, and across different conditions.

For these reasons, our current view is that using two orthogonal markers (mtDNAcn, CS) for the same construct (i.e., mitochondrial content) is likely to yield a more accurate estimate than either alone. This logic also constitutes part of the rationale for integrating these independent measurements into a single index (such as the MHI). Normalizing mitochondrial ETC activities to mitochondrial content is the gold standard to evaluate ETC dysfunction (Frazier et al., 2020), and can be directly compared to our MHI approach. For example, in mitochondrial disease where mtDNA-encoded ETC complexes I and IV activities can be severely impaired, we and others have observed dramatic, compensatory elevations in mitochondrial content (i.e., in muscle, 2-3-fold elevation in CS). The result is that the total tissue ETC activities can appear normal, but when normalized to mitochondrial content with CS, the ETC activity/CS ratio can be substantially reduced.

But as noted above, we cannot be certain that CS is the absolute ideal marker of mitochondrial content across all cell types. Therefore, there is added value to mathematically combine two orthogonal measures of the same construct into a single factor (e.g., CS + mtDNAcn for mitochondrial content; ETC complexes for global ETC activity). This approach has two main advantages: (i) it makes each construct more robust in the event that one of the metrics is sub-optimal (in a given cell type, for example) or that the rate limiting factor in ETC function depends on the combined activity of multiple enzymes, and also (ii) triangulates technical variation that is inherent to any laboratory measurement, making the resulting index more statistically robust.

Finally, MHI could be compared to other relevant aspects of mitochondrial function, for example using respirometry to assess maximal respiratory capacity. But again to obtain a measure of respiratory capacity on a per-mitochondrion basis, this measure would need to be expressed relative to mitochondrial content, which poses a similar challenge as with the MHI: with which energetic substrates do we assess respirometry, and which marker(s) do we use for mitochondrial content.

Figure 5: How do you define mitochondrial content when you have discrepant CS and mtDNAcn?

This is a great question. We define mitochondrial content as the volume within a cell that is occupied by mitochondria. While transmission electron microscopy (TEM) is considered the gold standard for measuring mitochondrial content, it is technically challenging to perform at scale. As mentioned above, although there is mixed evidence across different tissues, both CS and mtDNAcn remain positively correlated with mitochondrial content (e.g., (Larsen et al., 2012)). By integrating both markers into a single construct, we must obtain a more accurate estimate than either alone.

Figure 6: In here and other places, it is difficult to parse apart what is statistically significant and what are trends. An asterix (significance) appears in some places, but not in others where differences are being highlighted. Is figure 6g, mtDNAcn in CD8^+^ T-cells EM and CM really different for CS?

Thank you for this comment, we have revised the Results section to distinguish what is statistically significant and what is not. The majority of comparisons in Figure 6 are not significant because of the limited sample size. For this reason, we have focused our analysis and interpretation based on the standardized effect sizes (e.g., Hedge’s g), which is not systematically influenced by sample size. As requested, we have ensured that all statistical details are noted in the figure legend and that all statistical values denoted with asterisks are included.

Figure 10: Is this figure necessary? I am not sure it adds to your story. Are you trying to tell to many stories?

Figure 10 could certainly be removed. But as discussed above and as for Figure 6, our goal is to provide a resource and foundation for future studies. Figure 10 makes two points that have implications for the designing larger studies. First, mitochondrial features, only in some specific immune cell types, exhibit consistent patterns of correlation with some biomarkers (e.g., lipids). Second, using a cross-sectional cohort design (many individuals measured once) may yield different patterns of associations than a repeated-measures design. Again, these findings are enlightening and may provide much needed evidence to research groups seeking to select the best cell type, and the optimal study design, to address specific question.

Just overall, since you did do flow, I was surprised not see some flow measurements of mito parameters and correlation with other outcomes. However, I do appreciate the scale of this study and the work involved.

Our flow strategy had to be optimized for efficient cell sorting. On average, we flowed >100 million leukocytes to collect several 5 million cell aliquots from each of 6 cell types, per person. The sorting alone was a 6-hour procedure and required an optimized 14-color antibody panel. The addition of mitochondrial probes (which are notoriously less well behaved and specific than one would hope) would have introduced additional challenges and made cell sorting less efficient. For these reasons, we were not able to collect mitochondrial measurements from FACS.

Reviewer #2 (Recommendations for the authors):The methods identify the last author as the donor for the repeated measures analysis. I understand that the author consents but I'm not sure if this is consistent with eLife's policy with regard to human ethics, so I defer to that policy/practice but wanted to flag this. My sense is that it could be an undesirable precedent/practice to identify the biological/health data of authors, even with their consent, and it is unnecessary information for interpretation of the work.

Thank you for pointing out this potential concern. We have opted for the most transparent approach, consistent with previously published work in Nature Communications for example (e.g.,(Poldrack et al., 2015)). We defer to the editor for the most appropriate way to report this information in *eLife*.

Reviewer #3 (Recommendations for the authors):While the question is worthwhile asking, the sample size is too small to reach any conclusions due to the inherent heterogeneity of the group.

The study was primarily designed and powered to compare immune cell subtypes to each other. We acknowledge that the small sample size precludes stratification by sex and age, and therefore that some of the other findings are underpowered and should be considered exploratory. We have revised the manuscript to highlight these points.

The major observation is the discrepancy observed between the mitochondria phenotype of bulk PBMCs and isolated subsets and it is somewhat expected. Doing this in human is complicated due to the interindividual variation and sample size has to be much bigger to properly stratify the data among age groups, sexes, psychiatric disorders that the subjects may have, medications, lifestyle factors such as BMI, exercise, stress levels etc. In general, bulk PBMCs can never be accepted as a representative for the immune subsets even for mouse studies. In its original form, the data can serve as a resource for conducting assays related to mitochondrial profiling. But the hypothesis is not original and there are many confounding factors such as the heterogeneity of the cohort and insufficient sample size for the time-course study. While the findings may provide leads for future studies in human immune subsets, this reviewer doesn't recommend its publication in eLife.

Agreed. However, given that the target audience is not only the immunologist but also translational scientists who frequently use PBMCs out of convenience (because they are easier to obtain in high numbers than specific cell subtypes) we believe there is value in highlighting the discrepancy between bulk PBMCs and isolated cell subtypes and quantifying the magnitude of these mitochondrial differences in human immune cell subtypes.

Indeed, there is always a challenge in studying human subjects due to the natural inter-individual variability. However, researching humans is essential to define human health, normal physiology, and mechanisms of disease. While our study was exploratory and limited to a small sample size, we still have sufficient power to detect significant differences in mitochondrial parameters between subtype, which is the major finding of the study.

Additionally, we have revised the manuscript to highlight the need for larger human/clinical studies necessary to validate and build on these findings to effectively maximize our current understanding of human health and disease. In effort to capture the inherent heterogeneity of the cohort, we have obtained participant information including medical conditions, medications, and lifestyle factors such as BMI and exercise. Although we agree that our sample size is too small to stratify data across these groups, we hope future studies continue to build upon these data to ultimately include the stratification across these groups.

Overall, we hope that this serves as a foundation for future human/clinical studies in larger cohorts.

References:

Bindra, S., McGill, M. A., Triplett, M. K., Tyagi, A., Thaker, P. H., Dahmoush, L.,... Picard, M. (2021). Mitochondria in epithelial ovarian carcinoma exhibit abnormal phenotypes and blunted associations with biobehavioral factors. Scientific Reports, 11(1), 11595. doi:10.1038/s41598-021-89934-6

Frazier, A. E., Vincent, A. E., Turnbull, D. M., Thorburn, D. R., and Taylor, R. W. (2020). Assessment of mitochondrial respiratory chain enzymes in cells and tissues. Methods Cell Biol, 155, 121-156. doi:10.1016/bs.mcb.2019.11.007

Larsen, S., Nielsen, J., Hansen, C. N., Nielsen, L. B., Wibrand, F., Stride, N.,... Hey-Mogensen, M. (2012). Biomarkers of mitochondrial content in skeletal muscle of healthy young human subjects. The Journal of physiology, 590(14), 3349-3360. doi:10.1113/jphysiol.2012.230185

Márquez, E. J., Chung, C.-h., Marches, R., Rossi, R. J., Nehar-Belaid, D., Eroglu, A.,... Ucar, D. (2020). Sexual-dimorphism in human immune system aging. Nature Communications, 11(1), 751. doi:10.1038/s41467-020-14396-9

McLaughlin, K. L., Hagen, J. T., Coalson, H. S., Nelson, M. A. M., Kew, K. A., Wooten, A. R., and Fisher-Wellman, K. H. (2020). Novel approach to quantify mitochondrial content and intrinsic bioenergetic efficiency across organs. Scientific Reports, 10(1), 17599. doi:10.1038/s41598-020-74718-1

Picard, M. (2021). Blood mitochondrial DNA copy number: What are we counting? Mitochondrion, 60, 1-11. doi:https://doi.org/10.1016/j.mito.2021.06.010

Picard, M., Prather, A. A., Puterman, E., Cuillerier, A., Coccia, M., Aschbacher, K.,... Epel, E. S. (2018). A Mitochondrial Health Index Sensitive to Mood and Caregiving Stress. Biol Psychiatry, 84(1), 9-17. doi:10.1016/j.biopsych.2018.01.012

Poldrack, R. A., Laumann, T. O., Koyejo, O., Gregory, B., Hover, A., Chen, M.-Y.,... Mumford, J. A. (2015). Long-term neural and physiological phenotyping of a single human. Nature Communications, 6(1), 8885. doi:10.1038/ncomms9885

Rosenberg, A., Saggar, M., Rogu, P., Limoges, A. W., Sandi, C., Mosharov, E. V.,... Picard, M. (2021). Mouse brain-wide mitochondrial connectivity anchored in gene, brain and behavior. bioRxiv, 2021.2006.2002.446767. doi:10.1101/2021.06.02.446767

[Editors' note: further revisions were suggested prior to acceptance, as described below.]

The manuscript has been improved but there are some remaining issues that need to be addressed, as outlined below:While the reviewers were generally satisfied with the revisions, reviewers 2 and 3 and the reviewing editor feel there are issues that need to be addressed prior to acceptance.1. MHI. Reviewers 2 and 3 and the reviewing editor all find the added description and citations supporting the MHI to be insufficient to allay concerns over the lack of biological meaning, functional relevance, or interpretability of this measure. This analysis is not important to any of the key findings of this study, so we ask that Figure 5 and related text be moved to supplemental data.

Thank you for carefully reviewing our previous revisions. We are indebted to the reviewing editors and reviewers for helping to improve this manuscript well beyond the original version.

As requested, we have moved Figure 5 on MHI to supplemental data. This figure is now “Appendix 2-figure 1”. The related text also was removed from the main text, and moved to an appendix (Appendix 2).

2. Reviewers request further explicit acknowledgement that the distinct metabolic phenotypes for distinct cell populations is already well known in immunology. In your response, please clearly highlight text addressing this concern.

We have revised the manuscript in several places, to explicitly acknowledge the well-established metabolic differences across cell subtypes known in immunology. Examples from the main revised sections are shown below:

Abstract:

“These mitochondrial phenotyping data build upon established immunometabolic differences among leukocyte sub-populations and provide foundational quantitative knowledge to develop interpretable blood-based assays of mitochondrial health.”

Introduction:

“However, there are marked differences in the metabolic properties of different immune cell subtypes well known to immunologists. […] Thus, the immune system offers a well-defined landscape of metabolic profiles which, if properly mapped, can potentially serve as biomarkers.”

Discussion:

“Our biochemical and molecular results confirm and extend previous knowledge of bioenergetic differences across human immune cell types established using extracellular flux analysis, protein and mtDNA quantification (Chacko et al., 2013; Lee et al., 2021; Maianski et al., 2004; Pyle et al., 2010).”

“Thus, the observed cell type differences in mitochondrial content or RC activities across human immune cell subtypes likely reflect not only cellular ATP demand, but also the unique immunometabolic, catabolic/anabolic, and signaling requirements among different immune cell subtypes that contribute to produce cell-specific mitotypes.”

“Second, mitochondrial profiling precisely documented large-scale, quantitative differences in CS activity, mtDNAcn, and RC enzyme activities between well-known immune cell subtypes, contributing to our knowledge of the distinct metabolic characteristics among circulating immune cell types in humans. The qualitative and quantitative divergences were particularly emphasized by mitotypes, which highlighted conserved multivariate phenotypic features between lymphoid- and myeloid-derived immune cells, and between naïve and memory lymphocyte states.”

Reviewer #2 (Recommendations for the authors):The authors have responded to comments from the reviewers but I would have liked to have seen more engagement on some items.Specifically, two reviewers noted that they would have expected analysis of another more commonplace measure of mitochondrial metabolism alongside MHI. While I am aware of the short-comings of other methods, and cogniscient that it would be impractical to do at the same time as the sorting experiments that have been performed, a parallel analysis with a small subset of unsorted cells would be feasible. I view this kind of analysis as important to benchmark this MHI metric relative to other work in the field.All of the reviewers wanted to see more acknowledgement that the distinct metabolic phenotypes for distinct cells is well known in immunology and I see some changes but it is hard to gauge the extent of the changes. Also all of the reviewers wanted to see more tempered language for the statistical analyses and again I see some changes but hard to gauge. I can't see a track-changed copy in the documents but it would have been helpful to better assess how the language has changed.

As described above, we have revised the text in several places to acknowledge the well-established metabolic differences across cell subtypes well known in immunology. The MHI data and Figure 5 have also been moved to the supplement.

To address the comment around statistical analyses, we added p-values and adapted the language in several places to clarify which results are statistically significant and which are small effects or non-significant findings. Below are examples of the more tempered language highlighted:

Abstract

“Larger studies are required to validate and mechanistically extend these findings. These mitochondrial phenotyping data build upon established immunometabolic differences among leukocyte sub-populations, and provide foundational quantitative knowledge to develop interpretable blood-based assays of mitochondrial health.”

Results

“Given the exploratory nature of these analyses with a small sample size, only two results reached statistical significance (analyses non-adjusted for multiple testing).”

“Interestingly, across all cell subtypes examined, women showed a trend (p=0.0047, Chi-square) for higher CS activity […]”

“Other notable trends requiring validation in larger cohorts suggested that compared to women, men exhibited […]”

“In contrast, none of these potential differences were detectable in PBMCs, […]”

“Possibly due to the small sample size of our cohort, only CD4^+^ and CD8^+^ CM-EM T cells CII activity significantly increased with age (r=0.57 and 0.85, p<0.01 and 0.001 respectively, Figure 5j). However, CII activity tended to be positively correlated with age across all cell types except for monocytes and NK cells.”

“Overall, these exploratory data suggest that age-related changes in CS activity, mtDNAcn, and RC function are largely cell-type specific. […] Larger studies adequately powered to examine sex- and age-related associations are required to confirm and extend these results.”

Discussion

“Overall, our results provide standardized effect sizes of mitochondrial variation in relation to multiple key covariates, highlight the value of repeated-measures designs to carefully examine the mechanisms regulating mitochondrial health in humans, and call for the replication and extension of these findings in larger cohorts.”

“Sex and age analyses are exploratory and findings need to be validated by future adequately powered studies.”

The framing especially in the first couple of paragraphs of the introduction is still a bit unfocused and the challenges that the authors are aiming to address (validating an integrated measure of mitochondrial health for large translational studies, exploring the application of that measure with PBMCs that are a complex mix of cells) are not well described at the beginning of the abstract.

We have thoroughly revised the introduction to add clarity about the challenges addressed in this study.